# NEAT: NONLINEAR PARAMETER-EFFICIENT ADAPTATION OF PRE-TRAINED MODELS

## ABSTRACT

Fine-tuning pre-trained models is crucial for adapting large models to downstream tasks, often delivering state-of-the-art performance. However, fine-tuning all model parameters is resource-intensive and laborious, leading to the emergence of parameter-efficient fine-tuning (PEFT) methods. One widely adopted PEFT technique, Low-Rank Adaptation (LoRA), freezes the pre-trained model weights and introduces two low-rank matrices whose ranks are significantly smaller than the dimensions of the original weight matrices. This enables efficient fine-tuning by adjusting only a small number of parameters. Despite its efficiency, LoRA approximates weight updates using low-rank decomposition, which struggles to capture complex, non-linear components and efficient optimization trajectories. As a result, LoRA-based methods often exhibit a significant performance gap compared to full fine-tuning. Closing this gap requires higher ranks, which increases the number of parameters. To address these limitations, we propose a nonlinear parameter-efficient adaptation method (NEAT). NEAT introduces a lightweight neural network that takes pre-trained weights as input and learns a nonlinear transformation to approximate cumulative weight updates. These updates can be interpreted as functions of the corresponding pre-trained weights. The nonlinear approximation directly models the cumulative updates, effectively capturing complex and non-linear structures in the weight updates. Our theoretical analysis demonstrates that NEAT can be more efficient than LoRA while having equal expressivity. Extensive evaluations across four benchmarks and over twenty datasets demonstrate that NEAT significantly outperforms baselines in both vision and text tasks.

## 1 INTRODUCTION

Pre-trained models, trained on extensive and diverse general-domain corpora, demonstrate exceptional generalization capabilities, benefiting a range of fundamental tasks, such as natural language understanding (Devlin, 2018; Liu, 2019), natural language generation (Touvron et al., 2023a; AI@Meta, 2024), and image classification (Dosovitskiy et al., 2020a). In order to adapt pre-trained models to specific downstream tasks, fine-tuning is typically employed. However, due to the large number of parameters in pre-trained models, full fine-tuning requires significant computational resources and incurs substantial memory overhead (Qin et al., 2024).

To address this challenge, various parameter-efficient fine-tuning (PEFT) techniques (Ding et al., 2023; Han et al., 2024) have been developed, enabling pre-trained models to be fine-tuned in resource-constrained environments (Lin et al., 2024). PEFT methods reduce the memory overhead of fine-tuning by introducing a small set of learnable parameters, updating only these lightweight components. These approaches allow pre-trained models to effectively adapt to new tasks while minimizing resource consumption. Among PEFT techniques, the Low-Rank Adaptation (LoRA) family (Hu et al., 2021b; Liu et al., 2024; Song et al., 2024; Büyükakyüz, 2024; Zhao et al., 2024) is widely regarded as one of the most effective and popular approaches due to its minimal architectural modifications, high efficiency, and strong performance. The core concept of LoRA is to introduce low-rank matrices for each pre-trained model weight and approximate weight updates through their product. Since these low-rank matrices are much smaller than the original pre-trained weights, they significantly reduce the memory overhead during fine-tuning.

Despite its widespread success, LoRA has limitations, particularly in capturing complex relationships in weight updates. By decomposing weight updates into low-rank matrices, LoRA effectively reduces the fine-tuning parameter space, but this comes at the cost of failing to capture the non-linear interactions that are critical for many downstream tasks (Pan et al., 2024). This approximation often struggles to model the intricate optimization trajectories required for high performance, especially when the rank of the low-rank matrices is small. Consequently, LoRA-based methods often require higher ranks to close the gap between their performance and that of full fine-tuning, which increases the number of parameters and undermines the efficiency gains they were designed to achieve.

To overcome these limitations, we propose a novel nonlinear parameter-efficient adaptation method, NEAT, which incorporates a lightweight neural network into the adaptation process. Unlike LoRA, which approximates weight updates linearly through low-rank decomposition, NEAT models cumulative weight updates as functions of the pre-trained model's original weights. This enables NEAT to capture complex, non-linear patterns in the weight space, improving adaptation performance without increasing the number of parameters. The key innovation in NEAT lies in introducing a neural network that transforms the pre-trained weights, approximating the updates with minimal additional computation. This nonlinear transformation enhances the expressiveness of the parameter updates while maintaining the efficiency. Importantly, this architecture facilitates a more efficient exploration of the optimization landscape, leading to better task adaptation, particularly in cases where linear methods like LoRA would require much larger ranks to achieve competitive results. We theoretically demonstrate that NEAT can achieve the same or greater expressivity than LoRA with fewer parameters.

The contributions are summarized as follows:

- We propose NEAT, a novel parameter-efficient fine-tuning method that leverages nonlinear transformations, effectively capturing more complex weight updates. To the best of our knowledge, this is the first work to introduce nonlinear adaptation for LoRA-based PEFT methods.

- The proposed NEAT enhances model performance while maintaining the efficiency. We theoretically show that NEAT can achieve a possibly improved parameter efficiency compared to LoRA.

- We conduct extensive experiments on four benchmarks covering over twenty datasets. The experiments show that the proposed NEAT can outperform baselines on both vision and text tasks.

## 2 RELATED WORKS

In this section, we provide a concise overview of related work on Parameter-Efficient Fine-Tuning (PEFT) methods. PEFT methods aim to reduce the memory overhead of fine-tuning pre-trained models, enabling fine-tuning in resource-constrained environments. According to Han et al. (2024), PEFT methods can be categorized into: 1) **Additive PEFT methods** (Chronopoulou et al., 2023; Edalati et al., 2022; Lester et al., 2021; Wang et al., 2024c; Liu et al., 2022), 2) **Selective PEFT methods** (Guo et al., 2020; Das et al., 2023; Sung et al., 2021; Ansell et al., 2021; Zaken et al., 2021; Vucetic et al., 2022), 3) **Reparameterized PEFT methods** (Hu et al., 2021a; Valipour et al., 2022; Zhang et al., 2023; Karimi Mahabadi et al., 2021; Liu et al., 2024; Kopiczko et al., 2023), and 4) **Hybrid PEFT methods** (Mao et al., 2021; Chen et al., 2023; He et al., 2021; Zhang et al., 2022; Zhou et al., 2024). Among these, Low-Rank Adaptation (LoRA)-based methods, which are representative of reparameterized PEFT approaches, have gained significant attention due to their minimal architectural changes, no additional inference costs, and high efficiency. LoRA (Hu et al., 2021a) introduces two trainable low-rank matrices for each pre-trained model weight to approximate the desired updates of the original model. Extensions of LoRA include DyLoRA (Valipour et al., 2022), which dynamically adjusts the rank of the low-rank matrices during training to optimize for specific tasks; AdaLoRA (Zhang et al., 2023), which adaptively allocates the parameter budget among weight matrices based on their importance scores; and DoRA (Liu et al., 2024), which decomposes the pre-trained weight into magnitude and direction, applying LoRA only for direction updates. Other variants include VeRA (Kopiczko et al., 2023), which introduces shared frozen random matrices across layers to improve efficiency further, and RoseLoRA (Wang et al., 2024b), which employs a row- and column-wise sparse low-rank adaptation mechanism to selectively update the most significant parameters. FourierFT (Gao et al.) replaces the matrix multiplication in LoRA with a Fourier transform, while PiSSA (Meng et al., 2024) and MiLoRA (Wang et al., 2024a)

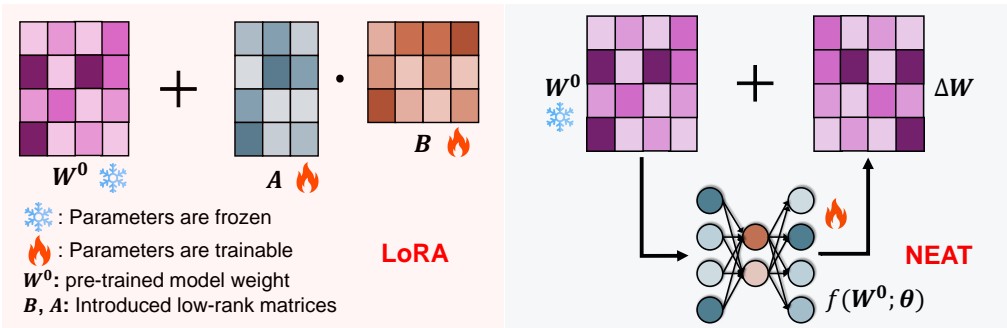

Figure 1: Framework of proposed NEAT.

update the principal and minor singular components of the weight matrix, respectively. However, existing PEFT methods rely on linear transformations to approximate pre-trained weight updates, which struggle to capture the complex relationships inherent in weight updates, leading to a significant performance gap compared to full fine-tuning. Meanwhile, existing research like (Teney et al., 2024) also demonstrates that nonlinear activation is an integral part of the neural network driving its success.

## 3 PRELIMINARY

In this section, we briefly introduce the preliminary of LoRA. LoRA assumes that the modifications to model weight matrices during fine-tuning exhibit low-rank properties. For a pre-trained weight matrix $\boldsymbol{W}^o \in \mathbb{R}^{d_1 \times d_2}$, LoRA models the efficient incremental update of pre-trained language models via the product of two learnable low-rank matrices

$$\boldsymbol{W} = \boldsymbol{W}^0 + \Delta \boldsymbol{W} = \boldsymbol{W}^0 + \boldsymbol{A}\boldsymbol{B}, \tag{1}$$

where $\boldsymbol{A} \in \mathbb{R}^{d_1 \times r}$ and $\boldsymbol{B} \in \mathbb{R}^{r \times d_2}$ with $r \ll \min(d_1, d_2)$.

During fine-tuning, only introduced two low-rank matrices $\boldsymbol{A}$ and $\boldsymbol{B}$ will be updated and the pre-trained weight $\boldsymbol{W}^0$ is frozen, which can be represented as

$$\min_{\boldsymbol{A}, \boldsymbol{B}} \mathcal{L}(\mathcal{D}_{\text{train}}; \boldsymbol{W}^o + \boldsymbol{A}\boldsymbol{B}), \tag{2}$$

where $\mathcal{D}_{\text{train}}$ is the training set used for fine-tuning and $\mathcal{L}$ is the loss function. Because $\boldsymbol{A}$ and $\boldsymbol{B}$ are two low-rank matrices and much more lightweight than $\boldsymbol{W}^0$, the LoRA costs much less memory space compared to the fully fine-tuning.

## 4 METHODOLOGY

### 4.1 FRAMEWORK OVERVIEW

As shown in Fig. 1, the proposed NEAT extends the incremental update mechanism of LoRA by introducing a non-linear weight adaptation approach for more expressive model updates. In LoRA, weight updates are achieved by decomposing adjustments into low-rank matrices $\boldsymbol{B}$ and $\boldsymbol{A}$, which are integrated into the pre-trained model weights $\boldsymbol{W}^0$. In contrast, NEAT enhances this by replacing the static low-rank updates with a dynamic method. Specifically, NEAT utilizes a neural network $f(\boldsymbol{W}^0; \boldsymbol{\theta})$ that takes the pre-trained weights $\boldsymbol{W}^0$ as input and generates a non-linear update $\Delta \boldsymbol{W}$. This design allows NEAT to capture more complex interactions and adapt more flexibly to a variety of tasks while maintaining parameter efficiency.

## 4.2 MOTIVATION

In fully fine-tuning of pre-trained models, the weight update process is typically performed through iterative gradient descent:

$$\boldsymbol{W}_t^0 = \boldsymbol{W}_{t-1}^0 - \eta \nabla_{\boldsymbol{W}_{t-1}^0} \mathcal{L} \tag{3}$$

where $\boldsymbol{W}_0^0 = \boldsymbol{W}^0$, $\eta$ is the learning rate, and $\boldsymbol{W}_t^0$ represents the weights after $t$ iterations. The cumulative change in the weights over time is represented as:

$$\Delta \boldsymbol{W} = \boldsymbol{W}_t^0 - \boldsymbol{W}_0^0. \tag{4}$$

This weight change $\Delta \boldsymbol{W}$ can be interpreted as a function of the original pre-trained weights $\boldsymbol{W}^0$, capturing the model's adaptation to the specific task during fine-tuning. Motivated by this observation, we propose to approximate $\Delta \boldsymbol{W}$ using a lightweight neural network that takes pre-trained model weight $\boldsymbol{W}^0$ as input and outputs the weight update directly. This approach leverages a non-linear network to model the weight updating directly, which can capture more complex and richer transformation of the weights efficiently.

## 4.3 NONLINEAR PARAMETER-EFFICIENT ADAPTATION

Similar to LoRA (Hu et al., 2021b), the proposed NEAT also provides incremental update of pre-trained language models. However, NEAT modifies the forward pass of the model by introducing a dynamic nonlinear weight transformation. Specifically, the model's forward propagation is formulated as:

$$\boldsymbol{y} = (\boldsymbol{W}^0 + f(\boldsymbol{W}^0; \boldsymbol{\theta}))\boldsymbol{x}, \tag{5}$$

where $\boldsymbol{x}$ and $\boldsymbol{y}$ are the input and output with respect to the current layer respectively and $f(\cdot; \boldsymbol{\theta}) : \mathbb{R}^{d_1 \times d_2} \to \mathbb{R}^{d_1 \times d_2}$ is a nonlinear neural network parameterized by $\boldsymbol{\theta}$. The neural network $f(\boldsymbol{W}^0; \boldsymbol{\theta})$ generates the weight update as a function of $\boldsymbol{W}^0$. In this formulation, the neural network $f(\boldsymbol{W}^0; \boldsymbol{\theta})$ allows for dynamic, non-linear weight updates that can capture more complex interactions than the static low-rank approximation used in standard LoRA. To ensure the efficiency of the proposed NEAT, the neural network $f(\boldsymbol{W}^0; \boldsymbol{\theta})$ should be lightweight, i.e., the number of parameters of $f(\boldsymbol{W}^0; \boldsymbol{\theta})$ is much smaller than that of original pre-trained weight $\boldsymbol{W}^0$. Therefore, we design the $f(\boldsymbol{W}^0; \boldsymbol{\theta})$ as a neural network with bottleneck layers. For example, a simple case is $f(\boldsymbol{W}^0; \boldsymbol{\theta}) = \sigma(\boldsymbol{W}^0 \boldsymbol{\Theta}_1) \boldsymbol{\Theta}_2$, where $\boldsymbol{\theta} = (\boldsymbol{\Theta}_1, \boldsymbol{\Theta}_2) \in \mathbb{R}^{d_2 \times r} \times \mathbb{R}^{r \times d_2}$ with $r \ll \min(d_1, d_2)$, and $\sigma(\cdot)$ is the activation function like ReLU (Glorot et al., 2011). We can also increase the layers or add activation function for the output of $f(\boldsymbol{W}^0; \boldsymbol{\theta})$ to enhance the model expressiveness.

During fine-tuning, the optimization objective is to minimize the task-specific loss function, which can be represented as

$$\min_{\boldsymbol{\theta}} \mathcal{L}(\mathcal{D}_{\text{train}}; \boldsymbol{W}^0 + f(\boldsymbol{W}^0; \boldsymbol{\theta})), \tag{6}$$

where the original pre-trained weight $\boldsymbol{W}^0$ is frozen, and only the parameters $\theta$ of the neural network $f(\boldsymbol{W}^0; \boldsymbol{\theta})$ are updated.

## 5 THEORETICAL ANALYSIS

In this section, we provide a theoretical analysis of the parameter efficiency of NEAT under ReLU activation. We first show that NEAT can achieve a similar expressivity than LoRA with possibly fewer parameters under certain conditions. Consider a design of shallow neural network $f(\boldsymbol{W}^0; \boldsymbol{\theta}) = \sigma(\boldsymbol{W}^0 \boldsymbol{\Theta}_1) \boldsymbol{\Theta}_2$ as in Section 4.3. Then, we have the following result about the expressivity of NEAT, where the expressivity is measured in terms of minimum attainable loss.

**Proposition 5.1.** *Let $\sigma$ be a ReLU activation function. Let $\boldsymbol{U}^0 \in \mathbb{R}^{d_1 \times \text{rank}(\boldsymbol{W}^0)}$ be the left singular vectors of $\boldsymbol{W}^0$. Suppose that the loss function for fine-tuning is invariant under the the projection of the weight matrix to the left singular space of $\boldsymbol{W}^0$, i.e., $\mathcal{L}(\mathcal{D}_{\text{train}}; \boldsymbol{W}) = \mathcal{L}(\mathcal{D}_{\text{train}}; \boldsymbol{U}^0 \boldsymbol{U}^{0\top} \boldsymbol{W})$ for*

*any $\boldsymbol{W} \in \mathbb{R}^{d_1 \times d_2}$. Then, for any $r \geq 1$,*

$$\min_{\substack{\boldsymbol{\Theta}_1 \in \mathbb{R}^{d_2 \times 2r}, \\ \boldsymbol{\Theta}_2 \in \mathbb{R}^{2r \times d_2}}} \mathcal{L}(\mathcal{D}_{train}; \boldsymbol{W}^0 + f(\boldsymbol{W}^0; (\boldsymbol{\Theta}_1, \boldsymbol{\Theta}_2))) \leq \min_{\substack{\boldsymbol{A} \in \mathbb{R}^{d_1 \times r}, \\ \boldsymbol{B} \in \mathbb{R}^{r \times d_2}}} \mathcal{L}(\mathcal{D}_{train}; \boldsymbol{W}^0 + \boldsymbol{A}\boldsymbol{B})$$

$$\leq \min_{\substack{\boldsymbol{\Theta}_1 \in \mathbb{R}^{d_2 \times r}, \\ \boldsymbol{\Theta}_2 \in \mathbb{R}^{r \times d_2}}} \mathcal{L}(\mathcal{D}_{train}; \boldsymbol{W}^0 + f(\boldsymbol{W}^0; (\boldsymbol{\Theta}_1, \boldsymbol{\Theta}_2))).$$

Proposition 5.1 demonstrates the (approximate) equivalence of LoRA and NEAT in terms of their expressivity. Specifically, the minimum attainable loss using rank-$r$ LoRA can be achieved by NEAT with $2r$ hidden units, and conversely, the minimum attainable loss using NEAT with $r$ hidden units can be achieved rank-$r$ LoRA, provided the invariance assumption holds. This equivalence further implies that the function classes realized by NEAT with $O(r)$ hidden dimensions and rank-$r$ LoRA are equivalent in expressivity, as the result holds for any loss functions.

Importantly, this highlights a potential improvement in parameter efficiency by NEAT. Namely, NEAT with $O(rd_2)$ parameters maintains the expressivity of LoRA with $r(d_1 + d_2)$ parameters. When $d_2 \ll d_1$, NEAT offers a significant improvement in parameter efficiency. In practice, $d_2 \ll d_1$ commonly appears in the first matrix of feed-forward layers of transformers (Vaswani, 2017; Dosovitskiy et al., 2021). In such cases, our theory suggests the improvement of NEAT over LoRA in parameter efficiency. The added parameter efficiency can also improve sample efficiency by allowing the model to learn representations with the same or fewer data points.

The invariance assumption in Proposition 5.1 pertains to the pre-trained model, and asserts that the later layers of the model depends solely on the task-relevant feature space. Given that we fine-tune a pre-trained model, the later layers are expected to capture this task-relevant feature space, which is described by the left singular space of $\boldsymbol{W}^0$. In practice, since the later layers primarily rely on this pre-trained feature space, the principal directions of the pre-trained weight matrix, represented by its singular vectors, encode most of the useful features for downstream tasks. This makes the loss largely invariant to changes outside this subspace. The proof is available in Section A.1.

If we consider a sinusoid activation function $\sigma_{\mathsf{p}}(x) = \sin(2\pi x)$, we can show a stronger result without the invariance assumption that NEAT has expressivity (almost) greater than or equal to a LoRA with possibly more parameters. We defer the result to the Appendix A.2.

## 6 EXPERIMENT

In the experiments, we evaluate the proposed NEAT and answer the following questions:

**RQ1** How does NEAT compare to state-of-the-art PEFT methods on NLP tasks?

**RQ2** How does NEAT compare to state-of-the-art PEFT methods on vision tasks?

**RQ3** How does the performance of NEAT vary with different fine-tuned modules, depths of the lightweight neural network, or non-linear activation functions?

### 6.1 DATASETS AND EXPERIMENT SETTINGS

#### 6.1.1 DATASETS

We conduct experiments on four different benchmarks:

- **Commonsense Reasoning**, including BoolQ (Clark et al., 2019), PIQA (Bisk et al., 2020), SocialIQA (Sap et al., 2019), HellaSwag (Zellers et al., 2019), WinoGrande (Sakaguchi et al., 2019), ARC-e, ARC-c (Clark et al., 2018) and OpenBookQA (Mihaylov et al., 2018) datasets, is formulated as multiple-choice problems. Following Wang et al. (2024a), we finetune LLaMA2-7B (Touvron et al., 2023a) and LLaMA3-8B (AI@Meta, 2024) on Commonsense170K (Hu et al., 2023) dataset which is a combined training dataset of these tasks, and evaluate the Accuracy on each test set.

- **Arithmetic Understanding** consists of two math reasoning datasets: GSM8K (Cobbe et al., 2021) and MATH (Hendrycks et al., 2021). We finetune LLaMA2-7B (Touvron et al., 2023a)

on MetaMath (Yu et al., 2023) dataset following Wang et al. (2024a). Models need to generate correct answers, and accuracy is used as the evaluation metric.

- **Natural Language Understanding** consists of eight datasets from the GLUE benchmark (Wang et al., 2018). We follow the evaluation metrics and setups from Gao et al. (2024); Wu et al. (2024b).

- **Image Classification** consists of eight datasets: OxfordPets (Parkhi et al., 2012), CI-FAR10 (Krizhevsky, 2009), DTD (Cimpoi et al., 2014), EuroSAT (Helber et al., 2019), RE-SISC45 (Cheng et al., 2017), StanfordCars (Krause et al., 2013), FGVC (Maji et al., 2013) and CIFAR100 (Krizhevsky, 2009) following Gao et al. (2024). The first five datasets have small label spaces, while the last three have large label spaces.

Further details on the datasets and hyper-parameters are provided in Appendix D and Appendix C respectively.

### 6.1.2 BASELINES

Our baselines are constructed on a task basis. Specifically, for each task, the proposed NEAT is compared with representative baselines from the corresponding domain, as listed below.

- For both Commonsense Reasoning and Arithmetic Understanding, following Wang et al. (2024a), LoRA (Hu et al., 2021b), PiSSA (Meng et al., 2024) and MiLoRA (Wang et al., 2024a) are employed as baselines. NEAT is applied to query, key, value, MLP up and MLP down layers.

- For Natural Language Understanding, we follow the setup from prior works (Gao et al., 2024; Wu et al., 2024b) that evaluate various representative PEFT methods, including LoRA (Hu et al., 2021b), Adapter Houlsby et al. (2019), BitFit (Zaken et al., 2021), RED (Wu et al., 2024a), DoRA (Liu et al., 2024), ReFT Wu et al. (2024b), and FourierFT (Gao et al., 2024).

- For Image Classification, we follow the setting of Gao et al. (2024) and take linear probing (LP), LoRA (Hu et al., 2021b) and FourierFT (Gao et al., 2024) as baselines. NEAT is applied to the query and value layers.

### 6.2 PERFORMANCE COMPARISON

### 6.2.1 COMMONSENSE REASONING

In this section, we present experiments on eight commonsense reasoning datasets to address RQ1, shown in Table 1. We compare the performance of three state-of-the-art baselines with the proposed NEAT across eight different datasets. NEAT consistently outperforms all baselines, achieving the highest accuracy on all tasks. Specifically, NEAT surpasses LoRA, PiSSA, and MiLoRA in terms of average accuracy by 4.6%, 10%, and 2.5%, respectively, using LLaMA2-7B as the backbone. Furthermore, when using LLaMA3-8B as the backbone, NEAT demonstrates average improvements of 4.9%, 11.8%, and 2.9% over LoRA, PiSSA, and MiLoRA, respectively. These results highlight the effectiveness and superiority of NEAT as a PEFT method.

### 6.2.2 ARITHMETIC REASONING

In this section, we present experiments on two arithmetic reasoning tasks, as shown in Table 2, to address RQ1. According to the table, full fine-tuning (FFT) achieves highest accuracy across the two datasets. However, the performance gap between the proposed NEAT and FFT is quite small, despite NEAT using significantly fewer trainable parameters. Moreover, compared to state-of-the-art PEFT baselines, the proposed NEAT achieves substantial performance improvements. In terms of average accuracy, NEAT demonstrates improvements of 7.5%, 12.4%, and 2.4% over LoRA, PiSSA, and MiLoRA, respectively. These results on arithmetic reasoning tasks suggest that NEAT is a highly effective and efficient fine-tuning method for complex reasoning tasks.

### 6.2.3 NATURAL LANGUAGE UNDERSTANDING

We conduct experiments on the GLUE to answer RQ1. The model performance is shown in Table 3. According to Table 3, the proposed NEAT significantly outperforms state-of-the-art PEFT methods.

Table 1: Accuracy comparison of LLaMA 2-7B (Touvron et al., 2023b) and LLaMA 3-8B (Dubey et al., 2024) against PEFT baselines on eight commonsense reasoning datasets. Results marked with "+" are taken from (Liu et al., 2024). Results marked with "*" are taken from (Wang et al., 2024a). The highest accuracy of methods per category are in **bold**. "AVG" means the average accuracy of all datasets.

| Model | PEFT | Accuracy (↑) | | | | | | | | |
|-------|------|-------|------|------|-----------|------------|-------|-------|------|------|
| | | BoolQ | PIQA | SIQA | HellaSwag | WinoGrande | ARC-e | ARC-c | OBQA | AVG |
| LLaMA2-7B | LoRA[+] | 69.8 | 79.9 | 79.5 | 83.6 | 82.6 | 79.8 | 64.7 | 81.0 | 77.6 |
| | PiSSA[*] | 67.6 | 78.1 | 78.4 | 76.6 | 78.0 | 75.8 | 60.2 | 75.6 | 73.8 |
| | MiLoRA[*] | 67.6 | 83.8 | 80.1 | 88.2 | 82.0 | 82.8 | 68.8 | 80.6 | 79.2 |
| | NEAT | **71.7** | **83.9** | **80.2** | **88.9** | **84.3** | **86.3** | **71.4** | **83.0** | **81.2** |
| LLaMA3-8B | LoRA[+] | 70.8 | 85.2 | 79.9 | 91.7 | 84.3 | 84.2 | 71.2 | 79.0 | 80.8 |
| | PiSSA[*] | 67.1 | 81.1 | 77.2 | 83.6 | 78.9 | 77.7 | 63.2 | 74.6 | 75.4 |
| | MiLoRA[*] | 68.8 | **86.7** | 77.2 | 92.9 | 85.6 | 86.8 | 75.5 | 81.8 | 81.9 |
| | NEAT | **71.9** | **86.7** | **80.9** | **94.1** | **86.7** | **90.9** | **78.7** | **84.4** | **84.3** |

Table 2: Accuracy comparison of LLaMA 2-7B against PEFT baselines on two arithmetic reasoning datasets. Results marked with "+" are taken from (Yu et al., 2023). Results marked with "*" are taken from (Wang et al., 2024a). The highest accuracy of methods per category are in **bold**. "AVG" means the average accuracy of all datasets.

| Method | GSM8K | MATH | AVG |
|--------|-------|------|-----|
| FFT [+] | 66.50 | 19.80 | 43.20 |
| LoRA[*] | 60.58 | 16.88 | 38.73 |
| PiSSA[*] | 58.23 | 15.84 | 37.04 |
| MiLoRA[*] | 63.53 | 17.76 | 40.65 |
| NEAT | **65.05** | **18.22** | **41.64** |

Specifically, NEAT-S, which uses a similar number of trainable parameters as FourierFT (Gao et al., 2024), DiReFT (Wu et al., 2024b), and LoReFT (Wu et al., 2024b), surpasses all PEFT baselines and experiences only a small performance drop (0.2%) compared to FFT. Additionally, NEAT-L exceeds the performance of all baselines, including FFT, with the same number of trainable parameters as LoRA. These results demonstrate that the proposed NEAT exhibits excellent generalization ability while maintaining high efficiency.

### 6.2.4 IMAGE CLASSIFICATION

In this section, we present the experiments on image classification datasets to address RQ2, shown in Table 4. From the table, NEAT significantly outperforms LoRA and FourierFT using the same number of trainable parameters. Specifically, NEAT achieves performance improvements of 11.05%, 7.30%, and 26.02% compared to LoRA, FourierFT, and LP, respectively. Furthermore, compared to FFT (86.49%), the proposed NEAT (86.15%) shows almost no performance drop while using only 0.3% of the trainable parameters required by FFT. This demonstrates that NEAT exhibits exceptional adaptation capability not only on NLP tasks but also on vision tasks. Additionally, it verifies the effectiveness of the nonlinear adaptation used in NEAT.

### 6.3 SENSITIVITY W.R.T. FINE-TUNED MODULE

In this section, we present the results of applying NEAT to various modules of ViT for image classification, addressing RQ3. The experimental results are shown in Fig. 3. We adjust the hidden layer dimension $r$ to maintain the same number of trainable parameters, ensuring a fair comparison. According to the figure, applying NEAT to the QV layers yields results similar to applying NEAT to both the QV and MLP layers. This indicates that NEAT is robust across different fine-tuning module selections, potentially reducing the need for extensive hyper-parameter tuning when applying NEAT to specific tasks.

Table 3: Accuracy comparison of RoBERTa-base against PEFT baselines on the GLUE benchmark. Baseline results with "*" are taken from Wu et al. (2024a). The highest accuracy of methods per category are in **bold**. "AVG" means the average accuracy of all datasets. NEAT-S refers to applying NEAT only to the layers starting from the 4th layer, with the hidden layer dimension of the neural network set to 1. This configuration matches the parameter count of FourierFT. In contrast, NEAT-L applies NEAT to all layers, with the hidden layer dimension set to 8, aligning the parameter budget with LoRA.

| PEFT | Params (%) | Accuracy (↑) | | | | | | | | |
|---|---|---|---|---|---|---|---|---|---|---|
| | | MNLI | SST-2 | MRPC | CoLA | QNLI | QQP | RTE | STS-B | AVG |
| FFT | 100% | 87.3 | 94.4 | 87.9 | 62.4 | 92.5 | 91.7 | 78.3 | 90.6 | 85.6 |
| Adapter* | 0.318% | 87.0 | 93.3 | 88.4 | 60.9 | 92.5 | **90.5** | 76.5 | 90.5 | 85.0 |
| LoRA* | 0.239% | 86.6 | 93.9 | 88.7 | 59.7 | 92.6 | 90.4 | 75.3 | 90.3 | 84.7 |
| Adapter$^{FNN}$* | 0.239% | **87.1** | 93.0 | 88.8 | 58.5 | 92.0 | 90.2 | 77.7 | 90.4 | 84.7 |
| BitFit* | 0.080% | 84.7 | 94.0 | 88.0 | 54.0 | 91.0 | 87.3 | 69.8 | 89.5 | 82.3 |
| RED* | 0.016% | 83.9 | 93.9 | 89.2 | 61.0 | 90.7 | 87.2 | 78.0 | 90.4 | 84.3 |
| FourierFT | 0.019% | 84.7 | 94.2 | 90.0 | 63.8 | 92.2 | 88.0 | 79.1 | 90.8 | 85.3 |
| DiReFT* | 0.015% | 82.5 | 92.6 | 88.3 | 58.6 | 91.3 | 86.4 | 76.4 | 89.3 | 83.2 |
| LoReFT* | 0.015% | 83.1 | 93.4 | 89.2 | 60.4 | 91.2 | 87.4 | **79.0** | 90.0 | 84.2 |
| NEAT-S | 0.019% | 84.9 | 94.3 | 90.2 | **64.6** | 92.0 | 88.3 | 78.3 | 90.5 | 85.4 |
| NEAT-L | 0.239% | 86.6 | **94.6** | 90.0 | 64.4 | **92.7** | 89.7 | 78.7 | **90.9** | **86.0** |

Table 4: Accuracy comparison of ViT-base (Dosovitskiy et al., 2020b) against PEFT baselines on the image classification benchmark. The reported accuracy (%) is obtained after 10 epochs. The highest accuracy of methods per category are in **bold**. "AVG" means the average accuracy of all datasets. Results marked with "*" are taken from Gao et al. (2024).

| Method | Params (M) | OxfordPets | StanfordCars | CIFAR10 | DTD | EuroSAT | FGVC | RESISC45 | CIFAR100 | AVG |
|---|---|---|---|---|---|---|---|---|---|---|
| FFT* | 85.8M | 93.14 | 79.78 | **98.92** | **77.68** | **99.05** | **54.84** | **96.13** | 92.38 | **86.49** |
| LP* | - | 90.28 | 25.76 | 96.41 | 69.77 | 88.72 | 17.44 | 74.22 | 84.28 | 68.36 |
| LoRA* | 581K | 93.19 | 45.38 | 98.78 | 74.95 | 98.44 | 25.16 | 92.70 | 92.02 | 77.58 |
| FourierFT* | 239K | 93.05 | 56.36 | 98.69 | 77.30 | 98.78 | 32.44 | 94.26 | 91.45 | 80.29 |
| NEAT | 258K | **93.77** | **80.03** | 98.70 | 77.57 | 98.79 | 53.60 | 94.27 | **92.47** | 86.15 |

## 6.4 SENSITIVITY W.R.T. DEPTH

As the depth of a neural network increases, the model gains more nonlinearity, potentially making NEAT more effective at capturing complex, non-linear patterns for weight updates. In this section, we present experiments with varying neural network depths in NEAT on the StanfordCars dataset to address RQ3, as shown in Fig. 2. The architecture of the stacked layers used in NEAT is shown in Fig. 5, with a detailed description provided in Appendix E. To ensure a fair comparison, we maintain consistent hyper-parameters across all configurations.

According to Fig. 2, increasing the network depth leads to better performance. Specifically, at a depth of 6 layers, the classification accuracy reaches 81.04%, marking a 1.7% improvement compared to using only 2 layers. When the depth is increased to 8 and 10 layers, the accuracy slightly decreases compared to the 6-layer model but remains higher than that of the 2-layer configuration. A possible explanation is that as depth increases—particularly at 10 layers—the training process becomes more challenging, possibly requiring more careful hyper-parameter tuning. It is also worth noting that, since the intermediate layers have much smaller dimensions ($\mathbb{R}^{r \times r}$ where $r$ is the hidden layer dimension) compared to the pre-trained model's weight dimensions, the additional parameter overhead of stacking more hidden layers is negligible and does not affect the parameter efficiency of NEAT. These results further demonstrate the effectiveness of introducing non-linear adaptation.

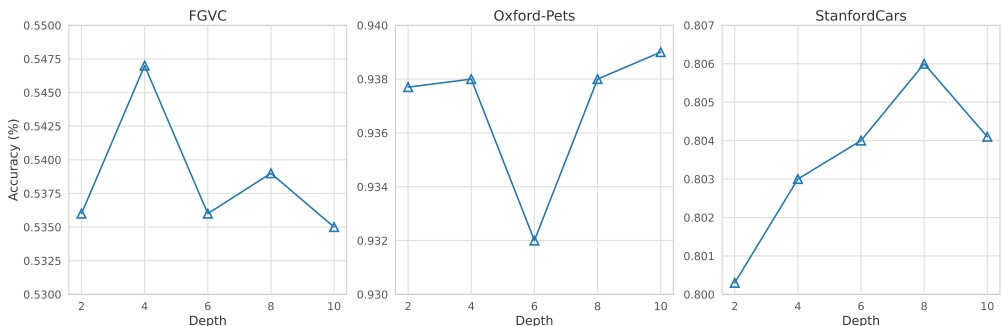

Figure 2: Accuracy on the StanfordCars, FGVC and Oxford-Pets dataset with varying depths of the neural network used in NEAT. The depth here represents the total number of layers in the neural network. We choose depth equals to 2, 4, 6, 8, and 10 layers in the figure.

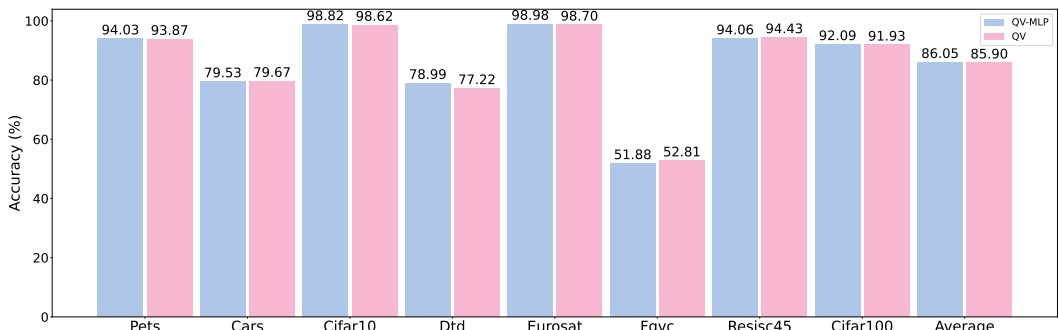

Figure 3: Accuracy of NEAT with different targeted fine-tuning modules, including just QV layers and a combination of QV and MLP layers, on image classification datasets.

Table 5: Accuracy of NEAT with different nonlinear activation functions, i.e. ReLU and sinusoid functions, on image classification datasets. The highest accuracy of methods per category are in **bold**. "AVG" means the average accuracy of all datasets.

| Method | OxfordPets | StanfordCars | CIFAR10 | DTD | EuroSAT | FGVC | RESISC45 | CIFAR100 | AVG |
|--------|-----------|--------------|---------|-----|---------|------|----------|----------|-----|
| ReLU | **93.87** | 79.67 | 98.62 | 77.22 | 98.70 | 52.81 | **94.43** | 91.93 | 85.90 |
| Sinusoid | 93.51 | **79.95** | **98.88** | **79.08** | **98.74** | **53.47** | 93.62 | **92.25** | **86.19** |

## 6.5 SENSITIVITY W.R.T. DIFFERENT NON-LINEAR ACTIVATIONS

A key innovation of NEAT compared to LoRA and other PEFT methods, which rely solely on linear transformations for modeling weight updates, is the introduction of non-linear activations within the adaptation neural network. Since the choice of non-linear activations directly influences the learning process and the dynamics of weight updates, we investigate the impact of different non-linear activations on overall adaptation performance to address RQ3. Specifically, we compare NEAT using $\sigma_p(x) = \sin(2\pi x)$ as the non-linear activation function with NEAT using ReLU, $\sigma_p(x) = \max(0, x)$. The results are presented in Table 5. To ensure a fair comparison, the number of trainable parameters remains the same, and hyperparameters such as learning rate are optimized to maximize performance. The specific hyper-parameters for the sinusoidal non-linear activation setting are provided in Appendix C.1.

According to the table, using a sinusoidal non-linear activation provides slightly better vision adaptation compared to ReLU. However, the performance gap is minimal, indicating that the choice of activation function does not significantly affect adaptation outcomes.

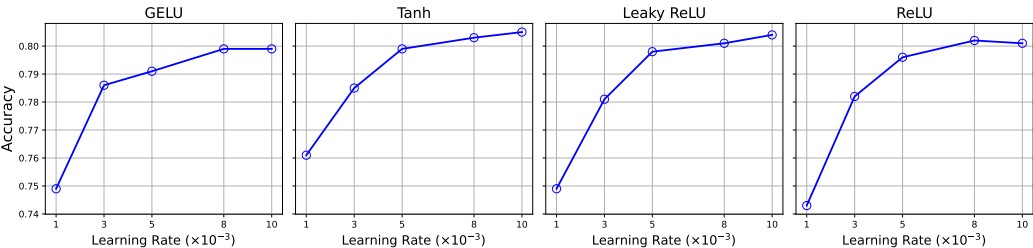

Figure 4: The performance of different nonlinear activations used in NEAT on the hyperparameter tuning. It can be observed that the pattern is mostly the same among all the nonlinear activations.

Table 6: Ablation results after running image classification datasets. The parameters count is the same and "AVG" means the average accuracy of all datasets.

| Method | OxfordPets | StanfordCars | CIFAR10 | DTD | EuroSAT | FGVC | RESISC45 | CIFAR100 | AVG |
|---|---|---|---|---|---|---|---|---|---|
| Nonlinear LoRA | 94.11 | 72.84 | 98.68 | 79.16 | 98.61 | 39.33 | 93.79 | 92.38 | 83.31 |
| Multiplicative LoRA | 93.57 | 77.32 | 98.68 | 77.57 | 98.81 | 46.79 | 94.34 | 91.86 | 84.81 |
| NEAT | 93.77 | 80.03 | 98.70 | 77.57 | 98.79 | 53.60 | 94.27 | 92.47 | 86.15 |

To further validate this observation and conduct a more detailed analysis of the influence of non-linear activations on hyperparameter tuning, we performed experiments on the StanfordCars dataset using various non-linear activation functions, including ReLU, Leaky ReLU, GELU, Tanh, and sinusoidal activation. These experiments involved varying learning rates for the adapters to analyze patterns in hyperparameter tuning across different activations. The results are illustrated in Fig. 4.

The findings reveal that, in general, the choice of activation functions does not necessitate specific hyperparameter tuning (e.g., learning rate). For instance, performance consistently improves with increasing learning rates, and the results for different activations remain comparable. This reinforces the conclusion that the choice of non-linear activations has a limited impact on overall adaptation performance. Consequently, ReLU can be a practical choice for achieving good adaptation results, particularly given its simplicity bias in neural networks, as highlighted in (Teney et al., 2024)

### 6.6 ABLATION STUDY

In this section, we present an ablation study with two variants of LoRA to demonstrate the effectiveness of our proposed framework: 1) nonlinear LoRA $y = (W_0 + \sigma(A)B)x$, and 2) multiplicative LoRA $y = (W_0 + W_0AB)x$. The experiments were conducted on image classification datasets, and the results are provided in Table 6. From the results, we observe that both nonlinear LoRA and multiplicative LoRA perform worse than NEAT. This highlights the effectiveness of incorporating nonlinear approximations and explicitly using model weights as input to the nonlinear function in our framework.

## 7 CONCLUSION

In this work, we propose NEAT, a novel parameter-efficient fine-tuning (PEFT) method that introduces nonlinear transformations to enhance model adaptation while maintaining efficiency. By incorporating a lightweight neural network that models cumulative weight updates as functions of the pre-trained weights, NEAT effectively captures complex, nonlinear structures in the weight space, allowing for more expressive and accurate adaptation to downstream tasks. Our theoretical analysis supports the efficacy of NEAT, demonstrating that it can achieve greater or equivalent expressiveness compared to existing LoRA, a popular and state-of-the-art PEFT method, with fewer number of parameters. Through extensive experiments on four benchmarks encompassing over twenty datasets with various pre-trained backbones, NEAT demonstrated superior performance on both NLP and vision tasks compared to existing state-of-the-art methods. NEAT thus stands out as an effective solution for fine-tuning pre-trained models more adaptively and efficiently, which is crucial for resource-constrained environments.

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

APPENDIX

# A DETAILS OF THEORETICAL RESULTS

In this section, we provide the proof of Proposition 5.1 and introduce additional theoretical results when we assume sinusoid activation.

## A.1 PROOF OF PROPOSITION 5.1

The intuition behind the proof is that we can always restore an identity function using two ReLU activation functions, i.e., $x = \sigma(x) - \sigma(-x)$ for any $x \in \mathbb{R}$

*Proof of Proposition 5.1.* We first show that

$$\min_{\boldsymbol{\Theta}_1 \in \mathbb{R}^{d_2 \times 2r}, \boldsymbol{\Theta}_2 \in \mathbb{R}^{2r \times d_2}} \mathcal{L}(\mathcal{D}_{\text{train}}; \boldsymbol{W}^0 + f(\boldsymbol{W}^0; (\boldsymbol{\Theta}_1, \boldsymbol{\Theta}_2))) \leq \min_{\boldsymbol{A} \in \mathbb{R}^{d_1 \times r}, \boldsymbol{B} \in \mathbb{R}^{r \times d_2}} \mathcal{L}(\mathcal{D}_{\text{train}}; \boldsymbol{W}^0 + \boldsymbol{A}\boldsymbol{B}).$$

Let $(\boldsymbol{A}^*, \boldsymbol{B}^*) = \arg\min_{\boldsymbol{A} \in \mathbb{R}^{d_1 \times r}, \boldsymbol{B} \in \mathbb{R}^{r \times d_2}} \mathcal{L}(\mathcal{D}_{\text{train}}; \boldsymbol{W}^0 + \boldsymbol{A}\boldsymbol{B})$. Take $\boldsymbol{\Theta}_1^{\#} :=$ $[(\boldsymbol{W}^0)^{\dagger} \boldsymbol{A}^*; -(\boldsymbol{W}^0)^{\dagger} \boldsymbol{A}^*] \in \mathbb{R}^{d_2 \times 2r}$ and $\boldsymbol{\Theta}_2^{\#} := [\boldsymbol{B}^{*\top}; -\boldsymbol{B}^{*\top}]^{\top} \in \mathbb{R}^{2r \times d_2}$, where $(\boldsymbol{W}^0)^{\dagger} \in \mathbb{R}^{d_2 \times d_1}$ is the Moore-Penrose inverse of $\boldsymbol{W}^0$. Then, since $\sigma$ is a ReLU activation function,

$$\begin{aligned} f(\boldsymbol{W}^0; (\boldsymbol{\Theta}_1^{\#}, \boldsymbol{\Theta}_2^{\#})) &= \sigma(\boldsymbol{W}^0 \boldsymbol{\Theta}_1^{\#}) \boldsymbol{\Theta}_2^{\#} \\ &= \sigma(\boldsymbol{W}^0 (\boldsymbol{W}^0)^{\dagger} \boldsymbol{A}^*) \boldsymbol{B}^* - \sigma(-\boldsymbol{W}^0 (\boldsymbol{W}^0)^{\dagger} \boldsymbol{A}^*) \boldsymbol{B}^* \\ &= \boldsymbol{W}^0 (\boldsymbol{W}^0)^{\dagger} \boldsymbol{A}^* \boldsymbol{B}^*. \end{aligned}$$

where the last equality follows since $x$ is in the column space of $\boldsymbol{W}^0$. Note that $\boldsymbol{W}^0 (\boldsymbol{W}^0)^{\dagger} = \boldsymbol{U}^0 \boldsymbol{U}^{0\top}$ is the projection to the left singular space of $\boldsymbol{W}^0$. Hence

$$\begin{aligned} \mathcal{L}(\mathcal{D}_{\text{train}}; \boldsymbol{W}^0 + f(\boldsymbol{W}^0; (\boldsymbol{\Theta}_1^{\#}, \boldsymbol{\Theta}_2^{\#}))) &= \mathcal{L}(\mathcal{D}_{\text{train}}; \boldsymbol{U}^0 \boldsymbol{U}^{0\top} \boldsymbol{W}^0 + \boldsymbol{U}^0 \boldsymbol{U}^{0\top} \boldsymbol{A}^* \boldsymbol{B}^*) \\ &= \mathcal{L}(\mathcal{D}_{\text{train}}; \boldsymbol{W}^0 + \boldsymbol{A}^* \boldsymbol{B}^*), \end{aligned}$$

where the last equality follows from the invariance assumption. This gives the first inequality:

$$\begin{aligned} \min_{\boldsymbol{\Theta}_1 \in \mathbb{R}^{d_2 \times 2r}, \boldsymbol{\Theta}_2 \in \mathbb{R}^{2r \times d_2}} \mathcal{L}(\mathcal{D}_{\text{train}}; \boldsymbol{W}^0 + f(\boldsymbol{W}^0; (\boldsymbol{\Theta}_1, \boldsymbol{\Theta}_2))) &\leq \mathcal{L}(\mathcal{D}_{\text{train}}; \boldsymbol{W}^0 + f(\boldsymbol{W}^0; (\boldsymbol{\Theta}_1^{\#}, \boldsymbol{\Theta}_2^{\#}))) \\ &= \mathcal{L}(\mathcal{D}_{\text{train}}; \boldsymbol{W}^0 + \boldsymbol{A}^* \boldsymbol{B}^*) \\ &= \min_{\boldsymbol{A} \in \mathbb{R}^{d_1 \times r}, \boldsymbol{B} \in \mathbb{R}^{r \times d_2}} \mathcal{L}(\mathcal{D}_{\text{train}}; \boldsymbol{W}^0 + \boldsymbol{A}\boldsymbol{B}). \end{aligned}$$

We next show the following inequality:

$$\min_{\boldsymbol{A} \in \mathbb{R}^{d_1 \times r}, \boldsymbol{B} \in \mathbb{R}^{r \times d_2}} \mathcal{L}(\mathcal{D}_{\text{train}}; \boldsymbol{W}^0 + \boldsymbol{A}\boldsymbol{B}) \leq \min_{\boldsymbol{\Theta}_1 \in \mathbb{R}^{d_2 \times r}, \boldsymbol{\Theta}_2 \in \mathbb{R}^{r \times d_2}} \mathcal{L}(\mathcal{D}_{\text{train}}; \boldsymbol{W}^0 + f(\boldsymbol{W}^0; (\boldsymbol{\Theta}_1, \boldsymbol{\Theta}_2))).$$

Take $\boldsymbol{A}^{\#} = \sigma(\boldsymbol{W}^0 \boldsymbol{\Theta}_1^*) \in \mathbb{R}^{d_1 \times r}$ and $\boldsymbol{B}^{\#} = \boldsymbol{\Theta}_2^* \in \mathbb{R}^{r \times d_2}$, where $(\boldsymbol{\Theta}_1^*, \boldsymbol{\Theta}_2^*) = \arg\min_{\boldsymbol{\Theta}_1 \in \mathbb{R}^{d_2 \times r}, \boldsymbol{\Theta}_2 \in \mathbb{R}^{r \times d_1}} \mathcal{L}(\mathcal{D}_{\text{train}}; \boldsymbol{W}^0 + f(\boldsymbol{W}^0; (\boldsymbol{\Theta}_1, \boldsymbol{\Theta}_2)))$. The conclusion follows from

$$\begin{aligned} \min_{\boldsymbol{A} \in \mathbb{R}^{d_1 \times r}, \boldsymbol{B} \in \mathbb{R}^{r \times d_2}} \mathcal{L}(\mathcal{D}_{\text{train}}; \boldsymbol{W}^0 + \boldsymbol{A}\boldsymbol{B}) &\leq \mathcal{L}(\mathcal{D}_{\text{train}}; \boldsymbol{W}^0 + \boldsymbol{A}^{\#} \boldsymbol{B}^{\#}) \\ &= \mathcal{L}(\mathcal{D}_{\text{train}}; \boldsymbol{W}^0 + \sigma(\boldsymbol{W}^0 \boldsymbol{\Theta}_1^*) \boldsymbol{\Theta}_2^*) \\ &= \min_{\boldsymbol{\Theta}_1 \in \mathbb{R}^{d_2 \times r}, \boldsymbol{\Theta}_2 \in \mathbb{R}^{r \times d_1}} \mathcal{L}(\mathcal{D}_{\text{train}}; \boldsymbol{W}^0 + f(\boldsymbol{W}^0; (\boldsymbol{\Theta}_1, \boldsymbol{\Theta}_2))). \end{aligned}$$

$\square$

## A.2 THEORETICAL ANALYSIS OF NEAT UNDER SINUSOID ACTIVATION FUNCTION

Here we consider a sinusoid activation function $\sigma_{\text{p}}(x) = \sin(2\pi x)$ (Gashler & Ashmore, 2014) and design $f(\boldsymbol{W}^0; \boldsymbol{\theta}) = \sigma_{\text{p}}(\boldsymbol{W}^0 \boldsymbol{\Theta}_1) \boldsymbol{\Theta}_2$ with $\boldsymbol{\theta} = (\boldsymbol{\Theta}_1, \boldsymbol{\Theta}_2)$. With this periodic activation function, we can show a stronger result that NEAT has expressivity (almost) greater than or equal to a LoRA with more parameters when $d_1 \gg d_2$.

**Proposition A.1** (Expressivity of NEAT with Sine Activation). *Suppose that there exists a row of $\boldsymbol{W}^0$, whose entries are linearly independent over the rationals. Then, for any $r > 0$, $\boldsymbol{A} \in \mathbb{R}^{d_1 \times r}$ and $\boldsymbol{B} \in \mathbb{R}^{r \times d_2}$, and $\epsilon > 0$, there exists some $\boldsymbol{\Theta}_1^* \in \mathbb{R}^{d_2 \times r}$ and $\boldsymbol{\Theta}_2^* \in \mathbb{R}^{r \times d_2}$ such that*

$$\|\boldsymbol{A}\boldsymbol{B} - \sigma_{\mathrm{p}}(\boldsymbol{W}^0\boldsymbol{\Theta}_1^*)\boldsymbol{\Theta}_2^*\|_{\mathrm{F}} \le \epsilon.$$

Proposition A.1 shows that the class of updates $\Delta\boldsymbol{W} = \sigma_{\mathrm{p}}(\boldsymbol{W}^0\boldsymbol{\Theta}_1)\boldsymbol{\Theta}_2$ by NEAT with $2rd_2$ parameters is dense in the class of updates $\Delta\boldsymbol{W} = \boldsymbol{A}\boldsymbol{B}$ by LoRA with $r(d_1 + d_2)$ parameters. When $d_2 \ll d_1$, this shows better parameter efficiency of NEAT.

Examining the proof of Proposition A.1, it is straightforward to show that the result holds for any continuous and periodic activation function whose range contains an open interval centered at 0.

*Proof.* This proof relies on Kronecker's theorem (Theorem 7.9 in Apostol (1990)) from number theory, which shows that for all $j \in \mathbb{R}^q$, the fractional parts of $(ct_1, ct_2, \ldots, ct_q)^\top$ is dense in $[0, 1]^q$ over $c \in \mathbb{R}$, as long as $t_1, \ldots, t_q$ are linearly independent over the rationals.

Let $\boldsymbol{w}_{j^*}$ be the $j^*$-th column of $\boldsymbol{W}^0$ whose entries are linearly independent over the rationals. Since $\boldsymbol{A}\boldsymbol{B}$ has a scale ambiguity, we can assume that $\boldsymbol{A}$ is a matrix whose entries are bounded by 1 without loss of generality. Write $\boldsymbol{A} = (\boldsymbol{a}_1, \boldsymbol{a}_2, \ldots, \boldsymbol{a}_r)$.

Take $\epsilon' > 0$ whose value will be determined later. From Kronecker's theorem, for each $\boldsymbol{a}_j$ there exists some $c_j \in \mathbb{R}$ such that

$$\left| \{c_j \boldsymbol{w}_{j^*}\} - \frac{\arcsin(\boldsymbol{a}_j)}{2\pi} \right| \le \epsilon',$$

where $\{\boldsymbol{b}\}$ is a vector whose entries are the fractional part of the corresponding entry of $\boldsymbol{b}$, and $\arcsin$ is applied elementwisely.

Let $\boldsymbol{\Theta}_1^* = (c_1 \boldsymbol{e}_{j^*}, c_2 \boldsymbol{e}_{j^*}, \ldots, c_r \boldsymbol{e}_{j^*})$, where $\boldsymbol{e}_{j^*}$ is the $j^*$-th standard basis vector in $\mathbb{R}^{d_2}$. Using the fact that $2\pi\{c_j \boldsymbol{w}_{j^*}\} = 2\pi c_j \boldsymbol{w}_{j^*} \mod 2\pi$, we have

$$\left\|\sigma_{\mathrm{p}}(\boldsymbol{W}^0\boldsymbol{\Theta}_1^*) - \boldsymbol{A}\right\|_{\mathrm{F}}^2 = \left\|\sigma_{\mathrm{p}}((c_1\boldsymbol{w}_{j^*}, c_2\boldsymbol{w}_{j^*}, \ldots c_r\boldsymbol{w}_{j^*})) - \boldsymbol{A}\right\|_{\mathrm{F}}^2 \tag{7}$$

$$\le \sum_j \|\sin(2\pi c_j \boldsymbol{w}_{j^*}) - \boldsymbol{a}_j\|^2 \le 4\pi^2 r\epsilon'^2, \tag{8}$$

where the last inequality follows from equation 8 and the fact that $\sin(x)$ is Lipschitz continuous with Lipschitz constant 1. Hence by choosing $\boldsymbol{\Theta}_2^* \leftarrow \boldsymbol{B}$, we have

$$\left\|\boldsymbol{A}\boldsymbol{B} - \sigma_{\mathrm{p}}(\boldsymbol{W}^0\boldsymbol{\Theta}_1^*)\boldsymbol{\Theta}_2^*\right\|_{\mathrm{F}}^2 \le \|\boldsymbol{B}\|^2 \left\|\sigma_{\mathrm{p}}(\boldsymbol{W}^0\boldsymbol{\Theta}_1^*) - \boldsymbol{A}\right\|_{\mathrm{F}}^2 \le 4\pi^2 \|\boldsymbol{B}\|^2 r\epsilon'^2.$$

Choose $\epsilon' = \epsilon/(2\pi\sqrt{r}\|\boldsymbol{B}\|)$, then the proof is complete. $\square$

# B  ADDITIONAL RELATED WORK

## B.1  ADDITIVE PEFT METHODS

Additive PEFT methods (Chronopoulou et al., 2023; Edalati et al., 2022; Lester et al., 2021; Wang et al., 2024c; Liu et al., 2022) introduces a small set of additional trainable parameters strategically placed within the model. One of the most prominent additive PEFT approaches is Adapter (Chronopoulou et al., 2023; Edalati et al., 2022; Zhao et al., 2022), which involves inserting small adapter layers between pre-trained weight blocks. Prompt Tuning (Wang et al., 2024c; Lester et al., 2021; Vu et al., 2021; Li & Liang, 2021) is another technique, where learnable vectors, or "soft prompts," are prepended to the input sequence without modifying the model's weights. This method is particularly effective for large-scale models and has inspired variants such as Prefix Tuning (Li & Liang, 2021).

## B.2 SELECTIVE PEFT METHODS

Selective PEFT focuses on optimizing the fine-tuning process by selectively adjusting a subset of the model's parameters rather than introducing additional ones. For instance, Diff Pruning (Guo et al., 2020) uses a learnable binary mask to select parameters for fine-tuning. Similarly, FishMask (Sung et al., 2021) and Fish-Dip (Das et al., 2023) leverage Fisher information to determine parameter importance and identify the most crucial ones for updates. Additionally, BitFit (Zaken et al., 2021) fine-tunes only the bias terms in the model, significantly reducing the number of trainable parameters.

## B.3 HYBRID PEFT METHOD

Hybrid PEFT methods aim to combine the strengths of various existing PEFT techniques to enhance model performance across diverse tasks. UniPELT (Mao et al., 2021) integrates LoRA, prefix-tuning, and adapters within each Transformer block, employing a gating mechanism to determine which module should be active during fine-tuning. S4 (Chen et al., 2023) further explores the design space by partitioning layers into groups and assigning different PEFT methods to each group. Additionally, NOAH (Zhang et al., 2022) and AUTOPEFT (Zhou et al., 2024) leverage neural architecture search (NAS) to automatically discover optimal combinations of PEFT techniques tailored to specific tasks.

## C HYPERPARAMETERS

We provide the specific hyperparameters used in our experiments to ensure reproducibility. For most of our experiments, we use the standard implementation of NEAT, which we refer to as vanilla NEAT. The neural network architecture in vanilla NEAT consists of only two layers: an input layer and an output layer. We selecte this approach because vanilla NEAT offers the benefits of simplicity in implementation, a low parameter count, and sufficient adaptation power. Nonetheless, we dedicate Section 6.4 and Appendix E to exploring more complex adaptation networks and their effect on performance.

### C.1 IMAGE CLASSIFICATION

Hyperparameters for NEAT in image classification are provided in Table 7. We tune the classification head and the backbone separately and provide detailed settings for each dataset. All weight decay values are not tuned and follow the settings from Gao et al. (2024). The scaling factor $s$ is set to 1.0. The hidden layer dimension $r$ for MHSA is set to 7 in the QV-setting, while both hidden layer dimensions for MHSA and MLP are set to 2 in the QV-MLP-setting described in Section 6.3. Additionally, specific hyper-parameters for the sinusoidal non-linear activation analysis are provided in Table 8.

Table 7: Hyperparameter of image classification for NEAT.

| Hyperparameter | OxfordPets | StanfordCars | CIFAR10 | DTD | EuroSAT | FGVC | RESISC45 | CIFAR100 |
|---|---|---|---|---|---|---|---|---|
| Epochs | | | | 10 | | | | |
| Optimizer | | | | AdamW | | | | |
| LR Schedule | | | | Linear | | | | |
| Weight Decay | 8E-4 | 4E-5 | 9E-5 | 7E-5 | 3E-4 | 7E-5 | 3E-4 | 1E-4 |
| QV | | | | | | | | |
| Learning Rate (NEAT) | 5E-3 | 1E-2 | 5E-3 | 1E-2 | 5E-3 | 1E-2 | 5E-3 | 5E-3 |
| Learning Rate (Head) | 5E-3 | 1E-2 | 5E-3 | 1E-2 | 5E-3 | 1E-2 | 1E-2 | 5E-3 |
| QV-MLP | | | | | | | | |
| Learning Rate (NEAT) | 5E-3 | 5E-3 | 5E-3 | 1E-2 | 5E-3 | 5E-3 | 1E-2 | 5E-3 |
| Learning Rate (Head) | 5E-3 | 1E-2 | 5E-3 | 1E-2 | 5E-3 | 1E-2 | 1E-2 | 5E-3 |

Table 8: Hyperparameters for image classification with NEAT using sinusoidal non-linear activation. The targeted modules are the same as the QV-setting (i.e., only adapting the query and value layers with a hidden layer dimension of 7).

| Hyperparameter | OxfordPets | StanfordCars | CIFAR10 | DTD | EuroSAT | FGVC | RESISC45 | CIFAR100 |
|---|---|---|---|---|---|---|---|---|
| Epochs | | | | 10 | | | | |
| Optimizer | | | | AdamW | | | | |
| LR Schedule | | | | Linear | | | | |
| Weight Decay | 8E-4 | 4E-5 | 9E-5 | 7E-5 | 3E-4 | 7E-5 | 3E-4 | 1E-4 |
| Learning Rate (NEAT) | 5E-3 | 5E-3 | 1E-3 | 5E-3 | 1E-3 | 5E-3 | 5E-3 | 1E-3 |
| Learning Rate (Head) | 5E-3 | 1E-2 | 5E-3 | 1E-2 | 5E-3 | 1E-2 | 1E-2 | 5E-3 |

Table 9: Hyperparameter of GLUE benchmark for NEAT-L.

| Hyperparameter | STS-B | RTE | MRPC | CoLA | SST-2 | QNLI | MNLI | QQP |
|---|---|---|---|---|---|---|---|---|
| Optimizer | | | | AdamW | | | | |
| LR Schedule | | | | Linear | | | | |
| Learning Rate (NEAT) | 5E-3 | 5E-3 | 5E-3 | 1E-3 | 5E-3 | 1E-3 | 5E-3 | 5E-3 |
| Learning Rate (Head) | 5E-3 | 5E-3 | 5E-3 | 1E-3 | 5E-3 | 1E-3 | 5E-3 | 5E-3 |
| Scaling | 0.1 | 0.01 | 0.01 | 0.1 | 0.01 | 0.01 | 0.01 | 0.01 |
| Max Seq. Len | 512 | 512 | 512 | 512 | 512 | 512 | 512 | 512 |
| Batch Size | 64 | 32 | 64 | 64 | 32 | 32 | 32 | 64 |

## C.2 NATURAL LANGUAGE UNDERSTANDING

We provide used hyper-parameters for NEAT in natural language understanding on the GLUE benchmark in Table 9 and Table 10. The learning rates for the head and the backbone are tuned separately. The scaling factor $s$ is searched in $\{0.01, 0.1, 1.0\}$. For reproducibility, we fix the seed as 0. The hidden layer dimension $r$ is set to 8 in NEAT-L and 1 in NEAT-S. More specifically, we apply NEAT to all layers in RoBERTa-base for NEAT-L, while only applying NEAT to layers $\{4, 5, 6, 7, 8, 9, 10, 11\}$ for NEAT-S to reduce the number of trainable parameters. The seed is fixed for reproducibility.

## C.3 COMMONSENSE REASONING

We provide hyperparameters settings of NEAT for commonsense reasoning task in Table 11. We follow the hyperparameters settings in MiLoRA (Wang et al., 2024a). We limit all samples to a maximum of 256 tokens. For evaluation, we set a maximum token number of 32.

## C.4 ARITHMETIC REASONING

We provide hyperparameters settings of NEAT for arithmetic reasoning task in Table 12. We follow the hyper-parameters settings in MiLoRA (Wang et al., 2024a). We limit all samples to a maximum of 2048 tokens. For evaluation, we set a maximum token number of 256 on GSM8K (Cobbe et al., 2021) dataset. On MATH (Hendrycks et al., 2021), we set the maximum new token to 512.

# D DATASETS

In this section, we provide a detailed description of the datasets used in our experiments.

## D.1 IMAGE CLASSIFICATION

For image classification, we provide detailed information about the used datasets in Table 13.

Table 10: Hyperparameter of GLUE benchmark for NEAT-S.

| Hyperparameter | STS-B | RTE | MRPC | CoLA | SST-2 | QNLI | MNLI | QQP |
|---|---|---|---|---|---|---|---|---|
| Optimizer | | | | AdamW | | | | |
| LR Schedule | | | | Linear | | | | |
| Learning Rate (NEAT) | 5E-3 | 1E-3 | 5E-3 | 5E-3 | 5E-3 | 1E-3 | 5E-3 | 1E-3 |
| Learning Rate (Head) | 1E-3 | 1E-3 | 5E-3 | 1E-3 | 5E-3 | 1E-3 | 5E-3 | 1E-3 |
| Scaling | 0.1 | 1.0 | 0.01 | 0.1 | 0.01 | 0.1 | 0.01 | 1.0 |
| Max Seq. Len | 512 | 512 | 512 | 512 | 512 | 512 | 512 | 512 |
| Batch Size | 64 | 32 | 64 | 64 | 32 | 32 | 32 | 64 |

Table 11: Hyperparameter of commonsense reasoning for NEAT.

| Hyperparameter | Commonsense Reasoning |
|---|---|
| Hidden Layer Dimension | 32 |
| $\alpha$ | 32 |
| Dropout | 0.05 |
| Optimizer | Adam W |
| Learning Rate | 3e-4 |
| Batch Size | 16 |
| Warmup Steps | 100 |
| Epochs | 1 |

## D.2 NATURAL LANGUAGE UNDERSTANDING

The GLUE benchmark comprises 8 NLP datasets: MNLI, SST-2, MRPC, CoLA, QNLI, QQP, RTE, and STS-B, covering tasks such as inference, sentiment analysis, paraphrase detection, linguistic acceptability, question-answering, and textual similarity. We provide detailed information about them in Table 14.

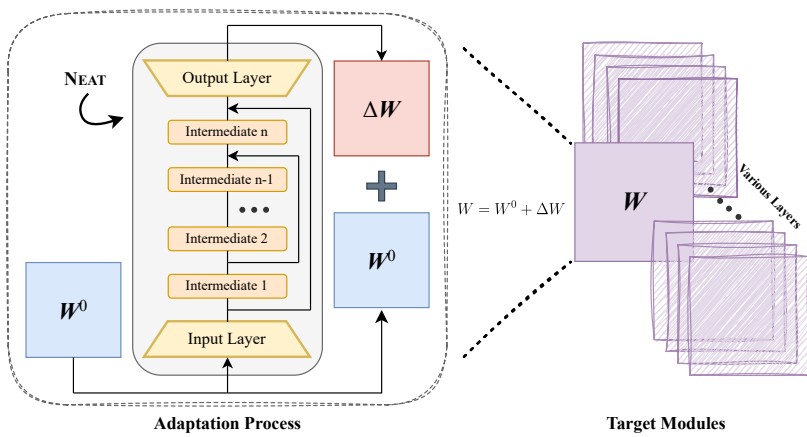

Figure 5: Implementation of introducing more depths to NEATt. We insert multiple intermediate layers into the layers from vanilla NEAT, with non-linear activation in between. The depth is described as the number of layers in NEAT, with vanilla NEAT having a depth of 2 (i.e. the input and output layers).

Table 12: Hyperparameter of arithmetic reasoning for NEAT.

| Hyperparameter | Arithmetic Reasoning |
|---|---|
| Hidden Layer Dimension | 64 |
| $\alpha$ | 64 |
| Dropout | 0.05 |
| Optimizer | Adam W |
| Learning Rate | 3e-4 |
| Batch Size | 16 |
| Warmup Steps | 100 |
| Epochs | 3 |

Table 13: Detailed information of image classification tasks.

| Dataset | #Class | #Train | #Val | #Test | Rescaled resolution |
|---|---|---|---|---|---|
| OxfordPets | 37 | 3,312 | 368 | 3,669 | |
| StandfordCars | 196 | 7,329 | 815 | 8,041 | |
| CIFAR10 | 10 | 45,000 | 5,000 | 10,000 | |
| DTD | 47 | 4,060 | 452 | 1,128 | $224 \times 224$ |
| EuroSAT | 10 | 16,200 | 5,400 | 5,400 | |
| FGVC | 100 | 3,000 | 334 | 3,333 | |
| RESISC45 | 45 | 18,900 | 6,300 | 6,300 | |
| CIFAR100 | 100 | 45,000 | 5,000 | 10,000 | |

Table 14: Detailed information of the GLUE benchmark. STS-B is a regression task, while all other tasks are either single-sentence or sentence-pair classification tasks.

| Corpus | Task | Metrics | # Train | # Val | # Test | # Labels |
|---|---|---|---|---|---|---|
| | | Single-Sentence Tasks | | | | |
| CoLA | Acceptability | Matthews Corr. | 8.55k | 1.04k | 1.06k | 2 |
| SST-2 | Sentiment | Accuracy | 67.3k | 872 | 1.82k | 2 |
| | | Similarity and Paraphrase Tasks | | | | |
| MRPC | Paraphrase | Accuracy/F1 | 3.67k | 408 | 1.73k | 2 |
| STS-B | Sentence similarity | Pearson/Spearman Corr. | 5.75k | 1.5k | 1.38k | 1 |
| QQP | Paraphrase | Accuracy/F1 | 364k | 40.4k | 391k | 2 |
| | | Inference Tasks | | | | |
| MNLI | NLI | Accuracy | 393k | 19.65k | 19.65k | 3 |
| QNLI | QA/NLI | Accuracy | 105k | 5.46k | 5.46k | 2 |
| RTE | NLI | Accuracy | 2.49k | 277 | 3k | 2 |

## D.3 COMMONSENSE REASONING

For commonsense reasoning task, we use 8 datasets, including BoolQ, PIQA, SIQA, HellaSwag, WinoGrande, ARC-e, ARC-c and OBQA. The detailed information is provided in Table 15.

## D.4 ARITHMETIC REASONING

Detailed information for arithmetic reasoning task is provided in Table 16. GSM8K consists of high quality grade school math problems, typically free-form answers. MATH includes classifi-

Table 15: Detailed information of commonsense reasoning task.

| Dataset | #Class | #Train | #Dev | #Test |
|---------|--------|--------|------|-------|
| BoolQ | Binary classification | 9,427 | 3,270 | 3,245 |
| PIQA | Binary classification | 16,113 | 1,838 | 3,000 |
| SIQA | Ternary classification | 33,410 | 1,954 | 2,224 |
| HellaSwag | Quaternary classification | 39,905 | 10,042 | 10,003 |
| WinoGrande | Binary classification | 40,398 | 1,267 | 1,767 |
| ARC-e | Quaternary classification | 2,251 | 570 | 2,376 |
| ARC-c | Quaternary classification | 1,119 | 229 | 1,172 |
| OBQA | Quaternary classification | 4,957 | 500 | 500 |

```python
class neat_depth_four(nn.Module):
    """
    Example of 4-layer implementation for Neat with residual.
    Using ReLU as the default non-linear activation function.
    args:
        dim: hidden dimension (a.k.a. rank)
        out_dim: output dimension
    """
    def __init__(self, dim=32, out_dim=768):
        super().__init__()
        self.non_linear = nn.ReLU()
        self.A = nn.Linear(out_dim, dim, bias=False)
        self.i1 = nn.Linear(dim, dim, bias=False)
        self.i2 = nn.Linear(dim, dim, bias=False)  # two intermediate layers
        self.B = nn.Linear(dim, out_dim, bias=False)
        nn.init.zeros_(self.B.weight)

    def forward(self, x, weight):
        delta_w = self.non_linear(weight @ self.A.weight.t())  # non-linear(W_0 A)
        residual = delta_w.clone()
        delta_w = self.non_linear(self.i1_(delta_w))
        delta_w = self.non_linear(self.i2_(delta_w))
        delta_w = delta_w + residual
        delta_w = self.B(delta_w)  # obtain the approximated delta W
        return x @ delta_w
```

Figure 6: An example of the actual implementation applying 4 layers in NEAT (depth = 4) with Pytorch.

Table 16: Detailed information of arithmetic reasoning task.

| Dataset | #Train | #Dev | #Test |
|---------|--------|------|-------|
| GSM8K | 7,473 | 1,319 | 1,319 |
| MATH | 12,500 | 500 | 5,000 |

cations from multiple mathematical domains, such as algebra, counting_and_probability, geometry, intermediate_algebra, number_theory, prealgebra and precalculus.

# E   IMPLEMENTATION OF INTRODUCING DEPTHS TO NEAT

We provide a comprehensive explanation of our approach to increasing the depth of the adaptation neural network in NEAT. As depicted in Fig. 5, we introduce multiple deeply stacked intermediate layers between the layers of the vanilla NEAT. These intermediate layers are essentially small adapters with a minimal parameter count ($\mathbb{R}^{r \times r}$, where $r$ is the hidden layer dimension), and we retain non-linear activations between them, as proposed by NEAT. The adaptation process begins by feeding the weight matrix $W^0$—the initialized value of the adaptation target $W$—into NEAT's input layer. After undergoing multiple non-linear transformations through the intermediate layers, the fi-

nal layer projects $W^0$ back to its original shape, producing the adaptation result $\Delta W$. Throughout this process, the adaptation target remains fixed, while all the intermediate layers, as well as the input and output layers in NEAT, are trainable parameters.

Furthermore, an implementation example of NEAT with four layers using the PyTorch library is illustrated in Fig. 6. As previously mentioned, we apply non-linear activations (ReLU in this case) to model more complex transformations. The intermediate layers have the same shape, $\mathbb{R}^{r \times r}$, which adds minimal overhead compared to $A \in \mathbb{R}^{d_2 \times r}$ and $B \in \mathbb{R}^{r \times d_2}$—the input and output layers, respectively, which are also present in the vanilla NEAT. Since $d_2$ is typically in the range of hundreds to thousands, while $r$ is commonly set to 8, 16, or 32, the parameter efficiency of NEAT with deeper layers remains comparable to that of vanilla NEAT without the intermediate layers. As shown, we first transform $W^0$ into the desired adaptation result $\Delta W$ and subsequently use $\Delta W$ to perform the actual computation on the input data. The use of residuals is based on empirical observations, as incorporating residual connections in the adaptation process results in faster convergence, more stable loss curves, and significantly improved overall performance.

