# OpenReview forum: "Neat: Nonlinear Parameter-efficient Adaptation of Pre-trained Models"
_ICLR.cc/2025/Conference — Submitted to ICLR 2025_

### Official Review · Reviewer_rriU · 2024-10-24

**Soundness:** 1
**Presentation:** 3
**Contribution:** 1
**Rating:** 3
**Confidence:** 5

**Summary:**

The paper introduces NEAT, a new method for fine-tuning pre-trained models efficiently by using nonlinear transformations. Traditional methods like Low-Rank Adaptation (LoRA) struggle to capture complex, nonlinear patterns in weight updates due to their linear approximation approach. NEAT overcomes this by incorporating a lightweight neural network that transforms pre-trained weights, allowing it to model more intricate updates without increasing the parameter count significantly. Theoretical analysis shows that NEAT is more expressive than LoRA with fewer parameters, and experiments demonstrate its superior performance across a wide range of vision and language tasks compared to existing parameter-efficient fine-tuning methods.

**Strengths:**

1. The paper is well-written and easy to follow.

2. The authors provide theoretical analysis to establish a lower bound for their method.

**Weaknesses:**

1. I have concerns about the paper's motivation, particularly the claim that LoRA struggles to capture complex, non-linear components and efficient optimization trajectories. In my opinion, LoRA models the differences in weight matrices, and the weight matrices are linear transformations. In neural networks, non-linearities are applied to activations, not the weight matrices themselves. Therefore, there is basically no motivation to add non-linearity to model the weight changes. Introducing non-linearity into the low-rank matrices A and B may contradict the core idea of "efficient fine-tuning" by adding unnecessary complexity.

2. The paper does not include comparisons of time or memory efficiency, which are crucial factors in the context of efficient fine-tuning methods. Including such evaluations would provide a more comprehensive understanding of the method's practicality.

3. The paper relies heavily on results from other studies, and I have concerns about their reproducibility and thus the fairness of these comparisons. It would be beneficial to introduce consistent experimental settings and perform fair comparisons to ensure the validity and reliability of the conclusions drawn.

**Questions:**

Please refer to the weakness part.

---

> ### Author Response · Authors · 2024-11-20
> **Rebuttal by Authors**
>
> >In neural networks, non-linearities are applied to activations, not the weight matrices themselves. Therefore, there is basically no motivation to add non-linearity to model the weight changes.
>
> Thank you for your feedback. We would like to clarify that the non-linearity is present in the model update. Equations (3) and (4) represent iterative model updates. The gradient of $L$ with respect to the model weight is a complex function of $W_0^{0}$, and the goal of PEFT is to approximate the weight update $\Delta W$. Although the transformer layers are nonlinear functions, they can not be used to capture the nonlinearity of the model update because they are frozen during fine-tuning. For the proposed NEAT, since $W_0^{0}$ is the input to this function, we aim to leverage a nonlinear network that takes $W_0^{0}$ as input to approximate the update directly. This approach allows us to efficiently capture more complex and richer transformations of the weights.
>
> >Introducing non-linearity into the low-rank matrices A and B may contradict the core idea of "efficient fine-tuning" by adding unnecessary complexity.
>
> In our simplest and most commonly applied settings, we use only the input and output layers with a single activation function (ReLU), without involving intermediate layers. This setup introduces minimal overhead to the runtime and memory usage of NEAT and does not impair its efficient fine-tuning capability. To support our claims, we report the runtime and memory consumption of NEAT and LoRA in Table 1 for the MRPC task. We ran 20 epochs with a parameter count of 0.3M, using other hyperparameters as specified in the paper.
> The results indicate that NEAT performs comparably to LoRA in terms of runtime and memory consumption (nearly identical), while achieving notably better accuracy. Additionally, we emphasize that NEAT introduces little additional complexity compared to LoRA in its basic form, which is the configuration primarily used in our experiments.
>
> Table 1: parameter efficiency in MRPC
> | method | param | time (sec)         | memory  |
> | ------ | ----- | ------------------ | ------- |
> | LoRA   | 0.3M  | 77.70872402191162s | 6916MiB |
> | NEAT   | 0.3M  | 78.4180045127868s  | 6916MiB |
>
> >include comparisons of time or memory efficiency.
>
> Thank you for pointing this out! In addition to the results shown in Table 1, we provide further efficiency analysis in Tables 2 and 3, using the COLA and STS-B datasets from the GLUE benchmark. We will include this analysis in the revised version of the paper.
> It can generally be observed that NEAT performs comparably to LoRA in terms of parameter efficiency. In most cases, NEAT consumes almost the same amount of memory as LoRA and has similar runtime performance. These observations demonstrate that NEAT is an efficient PEFT method with high parameter efficiency. Please note that due to different hyperparameter configurations across tasks, memory consumption may vary.
>
>
> Table 2: parameter efficiency in COLA
>
> | method | param | time (sec)       | memory  |
> | ------ | ----- | ---------------- | ------- |
> | LoRA   | 0.3M  | 66.7001121044159 | 3040MiB |
> | NEAT   | 0.3M  | 69.9160780906677 | 3040MiB |
>
>
> Table 3: parameter efficiency in STS-B
>
> | method | param | time (sec)        | memory  |
> | ------ | ----- | ----------------- | ------- |
> | LoRA   | 0.3M  | 117.9253075122833 | 7306MiB |
> | NEAT   | 0.3M  | 118.796623468399  | 7306MiB |
>
> >The paper relies heavily on results from other studies, and should reproduce them.
>
> This point certainly warrants further explanation. We conducted several pilot experiments using the open-source codebase provided by the authors and confirmed the reproducibility of their results before directly citing them. Additionally, we observed that some works are not open-source, and to ensure a fair comparison, we report their results as presented in their papers. To improve reproducibility, we will release our code in the final version.

---

> > ### Author Response · Authors · 2024-11-25
> >
> > Dear Reviewer rriU,
> >
> > Thank you for your time and insightful comments on our work. As we approach the end of the author-reviewer discussion period, we greatly value the opportunity to address your concerns.
> >
> > Could you kindly review our responses to see if they address your comments? We would highly appreciate it if you find our responses satisfactory and consider updating your rating. Feel free to reach out if you have any other questions or need further clarification.
> >
> > Best,
> >
> > Authors

---

> > ### Comment · Reviewer_rriU · 2024-11-26
> >
> > 1. I find the motivation unconvincing. The authors argue that iterative weight updates during fine-tuning introduce non-linearity, which the proposed method, Neat, is designed to address. However, the LoRA parameters $A$ and $B$ also require iterative updates, inherently introducing non-linearity (e.g., $A = \mathrm{optimize}(W_0^{0})$). This undermines the justification for the proposed motivation.
> >
> > 2. One of LoRA's strengths lies in its elegance, as it avoids the computational overhead of $\mathbb{R}^{d \times d} \times \mathbb{R}^{d}$ calculations. In contrast, Neat introduces this additional complexity.
> >
> > 3. The reported memory results seem questionable due to point 2. I believe a more detailed memory profiling or additional explanation is necessary to clarify this discrepancy.

---

> > > ### Author Response · Authors · 2024-11-26
> > >
> > > > I find the motivation unconvincing. The authors argue that iterative weight updates during fine-tuning introduce non-linearity, which the proposed method, Neat, is designed to address. However, the LoRA parameters ... This undermines the justification for the proposed motivation.
> > >
> > > We respectfully disagree with your opinion. First, the LoRA update is defined as $W+AB$, where $AB$ is a linear operation. The non-linearity in our method arises from the introduced nonlinear network, not from iterative updates. Could you clarify how the linear matrix product $AB$ introduces non-linearity? Additionally, could you explain the meaning of $A=optimize(W^{0}_0)$?
> > >
> > > > One of LoRA's strengths lies in its elegance, as it avoids the computational overhead ... I believe a more detailed memory profiling or additional explanation is necessary to clarify this discrepancy.
> > >
> > > Both LoRA and the proposed NEAT are parameter-efficient fine-tuning (PEFT) methods designed to reduce memory costs during fine-tuning. While training complexity is a consideration, it is not the primary focus of PEFT methods. For instance, well-known PEFT approaches such as AdaLoRA [1] and DoRA [2] introduce slight computational overhead but maintain parameter efficiency during fine-tuning. Similarly, NEAT incorporates a lightweight neural network, which introduces minimal computational overhead without increasing memory costs. As shown in Tables 1–3, we provide detailed comparisons of running time and memory consumption. Could you provide evidence supporting the claim that our proposed NEAT introduces significant computational overhead?
> > >
> > > [1] Zhang Q, Chen M, Bukharin A, et al. AdaLoRA: Adaptive budget allocation for parameter-efficient fine-tuning[J]. arXiv preprint arXiv:2303.10512, 2023.
> > >
> > > [2] Liu S Y, Wang C Y, Yin H, et al. Dora: Weight-decomposed low-rank adaptation[J]. arXiv preprint arXiv:2402.09353, 2024.

---

> ### Comment · Reviewer_rriU · 2024-11-27
>
> 1. Using the authors' formulation in Equation (2), we have $A = \mathrm{optimize}(W^0) = \mathrm{argmin} \mathcal{L}(\mathcal{D}_{\mathrm{train}}; W^0 + AB)$, which makes $A$ non-linear with respect to  $W^0$.
>
> 2. Your response is too high-level. Please consider me as someone without any prior knowledge of ML memory profiling. Could you provide a detailed memory profile for Neat? For example, specify the actual memory usage for storing pretrained model weights, activations, and gradients.

---

> > ### Author Response · Authors · 2024-11-27
> >
> > >Using the authors' formulation in Equation (2), we have … which makes $A$ non-linear with respect to $W^{0}$.
> >
> > LoRA introduces matrices $A$ and $B$ to approximate the model update $\Delta W$ using a linear approach. In contrast, our proposed method employs a neural network to approximate $\Delta W$, enabling a nonlinear approximation.
> >
> > >Your response is too high-level. Please consider me as someone without any prior knowledge of ML memory profiling. Could you provide a detailed memory profile for Neat? For example, specify the actual memory usage for storing pretrained model weights, activations, and gradients.
> >
> > Tables 1-3 present the time (in seconds) and memory costs of LoRA and NEAT during training. Memory cost refers to the total memory usage, including the storage of pretrained model weights, activations, gradients, and other necessary components. This total memory cost determines whether a model can be fine-tuned under limited resources. Due to the lack of tools to separately monitor memory usage for each component, we report the practical metric of total memory cost instead. For computational complexity, we measured the floating-point operations (FLOPs) of both LoRA and NEAT, which are identical at 92.50 GFLOPs. This demonstrates that NEAT does not introduce additional computational complexity compared to LoRA.

---

> > > ### Comment · Reviewer_rriU · 2024-11-28
> > >
> > > I have attempted to express my concerns to the authors, but they seem to reiterate their opinions rather than providing convincing evidence to address them. Given the unsatisfactory communication and my serious concerns about the paper's motivation, I feel compelled to advocate for its rejection.

---

### Official Review · Reviewer_fZ2J · 2024-11-03

**Soundness:** 3
**Presentation:** 3
**Contribution:** 3
**Rating:** 6
**Confidence:** 4

**Summary:**

Existing finetuning methods use linear adapters, either as an additive or multiplicative adaptation. However, this may be too restrictive of a function class to have as an adapter. Neat instead uses a small MLP to act as its adapter, with the architecture focused on in this paper being a 1-layer MLP. This has around the same number of parameters as LoRA for square parameter matrices, yet also contains an expressive nonlinearity. Empirically, this nonlinearity improves upon LoRA on multiple baselines. Theoretically, Neat is shown to be as expressive as LoRA depending on how the input and output dimensions compare, and when they are equal, Neat is at least as expressive as LoRA.

**Strengths:**

- The motivation is quite clear. Extending these fine-tuning methods to a non-linear adapter is quite natural.
- There is a theoretic understanding of the expressiveness of this method, both as an upper bound and a lower bound as compared with LoRA with varying parameter counts.
- The writing was clear, with both the method and any desired details easy to find through a cursory glance.
- Experiments into changing depth provide some insight into the behavior of changing the parameter count of Neat.

**Weaknesses:**

- For Proposition 5.1, there is an assumption that the loss function is invariant under projection to a left singular space. Listing some losses that are invariant under this projection would strengthen that result.
- For Figure 2 (and possibly other experiments), a single run for each depth doesn't provide strong confidence that this trend wasn't from random chance. A few more randomly initialized experiments to have some idea of the fluctuations over multiple runs would be useful.
- The experiments run are only for Neat. The other methods' accuracies come from other work rather than being reproduced, which may result in some differences simply from the training implementation.

**Questions:**

- Would it be possible to run the experiments with LoRA in your codebase? Having a comparison between the two methods under the training loop would make the benefit of Neat more convincing.
- In Table 3, the adaptation either starts at layer 4 with a single hidden dimension or is applied to all layers with 8 hidden dimensions. Is there any reason why these specific options were chosen?
- Can the parameter counts be reported in the experiments? At least for both Neat and LoRA to have that comparison.

Small things:
- In Tables 1 and 3, underlining the second-best accuracy would be nice.
- In Figure 2, can the depth 0 (i.e. base model) accuracies be added to the figure?

---

> ### Author Response · Authors · 2024-11-20
> **Rebuttal by Authors**
>
> >For Proposition 5.1, there is an assumption that the loss function is invariant under projection to a left singular space. Listing some losses that are invariant under this projection would strengthen that result.
>
> Thank you for the suggestion. We will add more details of the invariance assumption. Here is the intuition behind the assumption.
> The invariance assumption in Proposition 5.1 pertains to the _pre-trained model_ rather than the loss function itself. Specifically, the assumption posits that the loss, as a function of the parameter weights being fine-tuned, remains invariant under the projection $U U^\top$ onto the subspace spanned by the left singular vectors of the pre-trained weight matrix $W^0$.
> To illustrate this concept, consider a pre-trained model represented as a composition of three functions: $\mathcal{M}(x) = \mathcal{M}_3 \circ \mathcal{M}_2 \circ \mathcal{M}_1(x)$,
> where $\mathcal{M}_2(x) = W^0 x$ is the component being fine-tuned. The invariance assumption holds **if** the downstream function $\mathcal{M}_3$ satisfies the following property: it produces outputs of zero for any input that lies in the space orthogonal to the feature space of $\mathcal{M}_2$ (i.e., the subspace spanned by the left singular vectors of $W^0$).
> At a high level, this implies that $\mathcal{M}_3$ is specifically \textit{designed} or \textit{trained} to only "activate" in response to features produced by $\mathcal{M}_2$ during pre-training tasks. This assumption is reflective of how pre-trained models often encode meaningful structure within their dominant subspaces, while disregarding irrelevant directions.
> Furthermore, it is worth noting that this invariance assumption becomes unnecessary in certain cases, such as when sinusoidal activation functions are employed. In such settings, as shown in Proposition A.1 (Appendix), the output of the model inherently respects the structure of the feature space, rendering the assumption superfluous.
>
> >For Figure 2 (and possibly other experiments), a single run for each depth doesn't provide strong confidence that this trend wasn't from random chance.
>
> Thanks for your valuable suggestion. We run multiple runs for Figure 2 and report the results below.
>
> Table 1: multiple runs for multi-layer NEAT
> | depth | acc   |
> | ----- | ----- |
> | 4     | 0.803 |
> | 6     | 0.803 |
> | 8     | 0.806 |
> | 10    | 0.803 |
>
> The trend generally holds that introducing greater depth (i.e., intermediate layers) into the NEAT structure yields better results compared to vanilla NEAT.
>
> >The experiments run are only for Neat. The other methods' accuracies come from other work rather than being reproduced, which may result in some differences simply from the training implementation.
>
> Thank you for your comments. First, we use the exact same settings as the compared baselines, ensuring a fair comparison between our results and the baseline results. Additionally, some baselines did not release their code. To ensure a fair and convincing comparison, we directly report the results from their papers, which is a common practice for comparing experimental results in machine learning and NLP research.
>
> >Run lora in your codebase to provide a more convincing results.
>
> Thank you for the advice. We ran LoRA in our training loop and observed that the reproduced results are very close to those reported in the paper. Therefore, the comparison is fair.
>
> >In Table 3, the adaptation either starts at layer 4 with a single hidden dimension or is applied to all layers with 8 hidden dimensions. Is there any reason why these specific options were chosen? (answer : to match the parameter count.)
>
> Thank you for pointing this out! The former setup (NEAT-S) ensures a similar parameter count to FourierFT, which has the fewest parameters among the compared PEFT methods, allowing for a fair comparison. We began at layer 4 based on observations from AdaLoRA [2], which indicate that deeper layers are generally more important and allocating the parameter budget to these layers yields better results. Meanwhile, the latter setup (NEAT-L) ensures a similar parameter count to LoRA, which also uses a rank (hidden dimension) of 8.

---

> > ### Author Response · Authors · 2024-11-20
> > **Response to Reviewer fZ2J (part 2)**
> >
> > >Can the parameter counts be reported in the experiments? At least for both Neat and LoRA to have that comparison. (possibly refer to LLM experiments)
> >
> > Thank you for your suggestion. For all experiments in Section 6.2, we used the same number of parameters for NEAT and the baselines to ensure a fair comparison. For commonsense reasoning tasks, we show the trainable parameters, typical runtime, and computational overhead of the fine-tuning stage in Table 3. The results clearly show that the parameter volumes for both methods are identical. Our proposed method, NEAT, does not introduce any additional trainable parameters. This demonstrates that NEAT achieves comparable parameter efficiency to LoRA while delivering improved performance in the experiments. We will present the parameters used in Tables 1 and 2 in the revised version.
> >
> >
> > Table 3: parameter efficiency in Commonsense Reasoning:
> >
> > | method | params | time      | memory    |
> > | ------      | -----       | ---------- | -------        |
> > | LoRA    | 0.83%   | 5:35:10 | 23215MB |
> > | NEAT    | 0.83%   | 5:42:12 | 24447MB |
> >
> > >In Tables 1 and 3, underlining the second-best accuracy would be nice.
> >
> > Thanks for pointing this out! We will underline the second-best accuracy in our revised version.
> >
> > >In Figure 2, can the depth 0 (i.e. base model) accuracies be added to the figure?
> >
> > Thanks for your valuable comments. There may be a misunderstanding here, which we will clarify in the revised version.
> > If you are referring to the base model as vanilla NEAT, its depth is already 2 (consisting of the input layer and the output layer). The depth increases by adding intermediate layers between these two layers. Therefore, the accuracy for the base model in this scenario is presented in the figure.
> > If you are referring to the base model as the model without any adapters introduced (i.e., only the classifier is adapted), the accuracy is 25.76%. We will include this in the figure in the revised version.

---

> > > ### Comment · Reviewer_fZ2J · 2024-11-27
> > >
> > > Thank you for considering these suggestions! The intent was to show how much of an improvement NEAT (of any depth) provides over having no adapters, so a reader can visually measure how worthwhile fine-tuning with depth 4 is over fine-tuning with depth 2 (while taking fewer resources). However, after knowing the base model accuracy is different by an order of magnitude compared to the accuracy differences between depth 2 through 10, including that would be a hindrance, so the figure is great as is.

---

> > > > ### Author Response · Authors · 2024-11-27
> > > >
> > > > Thank you for your thoughtful and valuable feedback!

---

### Official Review · Reviewer_vgTy · 2024-11-04

**Soundness:** 1
**Presentation:** 2
**Contribution:** 1
**Rating:** 5
**Confidence:** 4

**Summary:**

This paper proposes NEAT to learn the adaptation matrices with the nonlinear transformation of the original weight matrices. NEAT applies an MLP to the pre-trained weight matrices and outputs their additive updates. Both theoretical and empirical analyses are provided to demonstrate the effectiveness of NEAT.

**Strengths:**

- The proposed method is simple.
- The authors have conducted extensive analysis of their proposed method.
- NEAT achieves promising results on both language and vision tasks.

**Weaknesses:**

- The authors argue that the main drawback of LoRA lies in the `low-rank` adaptation matrix. However, the employed multi-layer network in NEAT, which is $\sigma(\sigma(W^0 \theta_1)\cdots\theta_{l-1})\theta_l$, also leads to low-rank output. Let $\hat{A}=\sigma(\sigma(W^0 \theta_1)\cdots\theta_{l-1})$ and $B=\theta_l$. $\hat{A}$ can be regarded as the approximation of the matrix $A$ in LoRA. Given any $l\in\mathbb{N}$, and the same size $r$ where $A,\hat{A}\in\mathbb{R}^{d_1\times r}$ and $B\in\mathbb{R}^{r\times d_2}$, the rank
 of the learned adaptation matrix $AB$ or $\hat{A}B$ always equals $r$. Therefore, `NEAT fails to address the low-rank drawback of LoRA`.
- Given the high similarity between LoRA and NEAT, the authors did not provide sufficient effectiveness analysis. For example, why does the nonlinear approximation of LoRA perform better than the original LoRA? The authors should perform an ablation study on LoRA and NEAT given the same value of $r$ with different numbers of layers $l$. NEAT requires additional multi-layer neural networks to approximate the low-rank matrix. Therefore, what is the computational cost NEAT should take to outperform LoRA and the other PEFT methods?
- The proof of proposition 5.1 is not closed. It only covers the equality case; however, the strict inequality is not yet established.
- The author should provide the parameter volumes for the methods in Tab 1-2 for a fair comparison.
- More datasets should be included in Figure 2. Results on a single dataset are less convincing and can not give rise to persuasive conclusions.

**Questions:**

Please refer to weaknesses.

---

> ### Author Response · Authors · 2024-11-20
> **Rebuttal by Authors**
>
> >the employed multi-layer network in NEAT, which is σ(σ(W0θ1)⋯θl−1)θl, also leads to low-rank output.
>
> Thank you for your valuable question. Our contribution lies in proposing a nonlinear parameter-efficient fine-tuning framework. The benefit comes from the nonlinear approximation of model updates rather than high-rank approximation. We find that the model update is a complex function of $W^{0}$. Therefore, we introduce a nonlinear network that takes $W^{0}$ as input to approximate the update, providing an explicit method to achieve this. This enables NEAT to capture complex, nonlinear patterns in the weight space, thereby improving adaptation performance without increasing the number of parameters. Importantly, this architecture facilitates more efficient exploration of the optimization landscape, leading to better task adaptation, particularly in cases where linear methods like LoRA would require much larger ranks to achieve competitive results. We theoretically demonstrate that NEAT can achieve equal or greater expressivity compared to LoRA with fewer parameters. Furthermore, under our general framework, it also can be straightforward to achieve a high-rank update: $\sigma(\sigma(W^{0}\Theta_{1})\Theta_{2})$. However, we find that without the first activation function, NEAT has outperformed all baseline methods.
>
> >the authors did not provide sufficient effectiveness analysis
>
> Thanks for your feedback. Here are some responses to your concerns:
>
> 1. Why does the nonlinear approximation of LoRA perform better than the original LoRA? This is analyzed from both theoretical and empirical perspectives in the paper. NEAT can model more complex weight updates, leading to a more expressive adaptation process.
>
> 2. The authors should perform an ablation study on LoRA and NEAT given the same value of r with different numbers of layers. It is important to note that in our experiments, we use the same value of $r$ for both NEAT and LoRA, resulting in the same parameter count (using vanilla NEAT). Therefore, our comparison is fair.
>
> Table 1: parameter efficiency in MRPC
> | method | param | time (sec)         | memory  |
> | ------ | ----- | ------------------ | ------- |
> | LoRA   | 0.3M  | 77.70872402191162s | 6916MiB |
> | NEAT   | 0.3M  | 78.4180045127868s  | 6916MiB |
>
> >What is the computational cost NEAT should take to outperform LoRA and the other PEFT methods?
>
> Thanks for pointing out the lack of computational cost analysis. First thing we want to note is that **NEAT does not need the multiple intermediate layers to outperform LoRA**. In fact, we use the vanilla NEAT in most parts of our experiments (besides some analysis).
>
> Here we analyze the computational cost of vanilla NEAT (**actual implementation we use for experiments!**) and LoRA, running MRPC, STS-B tasks from GLUE benchmark and the fine-tuning stage of commonsense reasoning tasks. **Kindly note that due to the different hyperparameter configuration across various tasks, the memory consumption varies across tasks.**
>
>
> Table2: hardware efficiency in MRPC:
>
>
> | method | param | time (sec)         | memory  |
> | ------ | ----- | ------------------ | ------- |
> | LoRA   | 0.3M  | 77.70872402191162s | 6916MiB |
> | NEAT   | 0.3M  | 78.4180045127868s  | 6916MiB |
>
>
> Table 3: hardware efficiency in STS-B:
>
> | method | param | time (sec)        | memory  |
> | ------ | ----- | ----------------- | ------- |
> | LoRA   | 0.3M  | 117.9253075122833 | 7306MiB |
> | NEAT   | 0.3M  | 118.796623468399  | 7306MiB |
>
>
> Table 4: hardware efficiency in Commonsense Reasoning:
>
> | method | params | time      | memory    |
> | ------      | -----       | ---------- | -------        |
> | LoRA    | 0.83%   | 5:35:10 | 23215MB |
> | NEAT    | 0.83%   | 5:42:12 | 24447MB |
>
>
> It can be observed that the implementation we use for the experiments of NEAT matches the hardware efficiency of LoRA, with similar runtime and memory consumption (almost identical). For your convenience, we also include the data from multi-layer with 2 intermediate layers of NEAT in MRPC task below.
>
>
> Table 5: hardware efficiency of multi-layer NEAT
> | param | time (sec)         | memory  | dataset |
> | ----- | ------------------ | ------- | ------- |
> | 0.3M  | 92.91527390480042  | 3436MiB | cola    |
> | 0.3M  | 1355.3571300506592 | 2896MiB | sst-2   |
>
>
> It can further be noticed that even when introducing intermediate layers to form a small-scale neural network, the computational cost of NEAT is still comparable to LoRA, while achieving superior performance for the task!

---

> ### Author Response · Authors · 2024-11-20
> **Response to Reviewer vgTy (part 2)**
>
> >The proof of proposition 5.1 is not closed. It only covers the equality case; however, the strict inequality is not yet established.
>
> Thank you for your feedback. We have added additional details to complete the proof of Proposition 5.1 in the appendix.
> The key message of Proposition 5.1 lies in demonstrating the equivalence of **expressivity** between LoRA and NEAT. Specifically, our goal is to establish that both methods can achieve comparable levels of representation power under the assumptions outlined in the proposition, while NEAT may require fewer parameters. Thus, the focus is on equivalence rather than demonstrating a strict inequality between the two approaches.
>
> >Provide the parameter volumes (count) for the methods in Tab 1-2 for a fair comparison.
>
> Thanks for pointing this out! We provide the parameter count in the LLM experiments below.
>
> For commonsense reasoning tasks, we provide trainable parameters, typical runtime and computational overhead of the fine-tuning stage below.
>
> Table 6: parameter efficiency of NEAT in Commonsense Reasoning:
>
> | method | params | time      | memory    |
> | ------      | -----       | ---------- | -------        |
> | LoRA    | 0.83%   | 5:35:10 | 23215MB |
> | NEAT    | 0.83%   | 5:42:12 | 24447MB |
>
> It is evident that the parameter volumes for both methods are identical. Our proposed method, NEAT, does not introduce any additional trainable parameters. This ensures that NEAT maintains the same efficiency in terms of parameter count as LoRA, while achieving superior performance in the experiments.
>
> >One dataset is not enough for the study of depth. (Figure 2)
>
> Thanks for your advice, we run this analysis on FGVC and DTD tasks as well to further illustrate our point! We will add the result in the revised version of the paper.
>
> Table 7: multi-layer NEAT in FGVC
>
> | depth | acc   |
> | ----- | ----- |
> | 4     | 0.547 |
> | 6     | 0.536 |
> | 8     | 0.539 |
> | 10    | 0.535 |
>
>
> Table 8: multi-layer NEAT in Oxford-Pets
>
> | depth | acc   |
> | ----- | ----- |
> | 4     | 0.938 |
> | 6     | 0.932 |
> | 8     | 0.938 |
> | 10    | 0.939 |
>
>
> These results essentially demonstrate that the same phenomenon can generalize well to other tasks and is not a result of randomness. Note that increasing the depth will not always bring performance gain and this is also discussed in our paper.

---

> ### Comment · Reviewer_vgTy · 2024-11-22
>
> Thanks for the response. Based on the author responses and the reviews from the other reviewers, my main concerns now lie in the comparison between LoRA and NEAT.
>
> Given the formulation of LoRA as $W=\Delta W + AB$, the key point of NEAT is to approximate the low-rank matrix $A$ with the non-linear transformation of $\Delta W$. However, the authors have not provided sufficient supporting arguments on why NEAT can outperform LoRA. More actual justifications should be provided instead of 'efficient exploration of the optimization landscape' and 'capture complex, nonlinear patterns in the weight space'.
> - Theoretically in proposition 5.1, under the same value of $r$, NEAT is not guaranteed to outperform LoRA. The proof of proposition 5.1 is still not closed in the revision. The authors only demonstrate the conditional equivalence between LoRA and NEAT, which can not prove the superiority of NEAT. The proved conditional equivalence of NEAT and LoRA is the minimum expectation. They share the same adaptation structure $AB$, which ensures the same computational requirements during inference. However, NEAT introduces additional computational overhead by approximating $A$ during fine-tuning. Therefore, the authors must provide stronger evidence to demonstrate that NEAT is indeed superior to LoRA. Without this justification, it is unclear why the additional complexity introduced by NEAT is warranted.
> - Empirically, careful tuning for all baselines should be conducted. Also, the resource requirement during fine-tuning is important. I noticed that the authors sometimes compare LoRA and NEAT in the inference stage (same FLOPs, same number of parameters, etc.). Moreover, the ablation study on depth varies across different datasets, so the authors should repeat their experiments to improve the statistical reliability of the results.
>
> I appreciate the authors' efforts during the rebuttal to address my concerns. However, given the current lack of proof (which would require further peer review), the need for tuning the baselines, and other issues, the manuscript still requires substantial revisions and does not yet meet the standards for acceptance.

---

> > ### Author Response · Authors · 2024-11-22
> > **Response to Reviewer vgTy (Second Round)**
> >
> > >Theoretically in proposition 5.1, under the same value of $r$, NEAT is not guaranteed to outperform LoRA. The proof of proposition 5.1 is still not closed in the revision. The authors only demonstrate the conditional equivalence between LoRA and NEAT, which can not prove the superiority of NEAT. The proved conditional equivalence of NEAT and LoRA is the minimum expectation. Therefore, the authors must provide stronger evidence to demonstrate that NEAT is indeed superior to LoRA. Without this justification, it is unclear why the additional complexity introduced by NEAT is warranted.
> >
> > We appreciate the reviewer’s comment and would like to clarify that Proposition 5.1 establishes the equivalence between LoRA and NEAT in terms of expressivity. At the same time, NEAT requires fewer parameters to achieve the same level of expressivity when $d_2 \ll d_1$, a scenario that commonly arises in the first weight matrix of feed-forward layers [4,5]. This result highlights that NEAT is more parameter-efficient compared to LoRA. Thus, the proof of Proposition 5.1 is complete to substantiate its claim.
> > More specifically, the argument in Section 5 consists of 2 parts:
> > 1. Equivalence under the same order of hidden dimensions (low-rankness) in Proposition 5.1.
> > 2. Parameter efficiency of NEAT. The discussion following Proposition 5.1 demonstrates that NEAT can enjoy improved **parameter efficiency** in scenarios where $d_2 \ll d_1$. This scenario happens in the second matrix of feed-forward layers in transformers.
> >
> > Additionally, we add more explanations to justify the invariance assumption, which posits that the later layers of pre-trained models depend solely on the feature space of the weight matrix during fine-tuning (see also our response to Reviewer fZ2J). Moreover, we note that this assumption can be removed by using a sine activation function instead of ReLU. With this adjustment, we can remove the invariance assumption entirely and formally establish that the expressivity of NEAT is greater than or equal to that of LoRA.
> > In light of your comments, we have updated these sections in the revised manuscript to improve clarity and make the logical progression of our argument more explicit.
> > We hope these revisions and clarifications address the reviewer’s concerns.
> >
> > >Empirically, careful tuning for all baselines should be conducted. Also, the resource requirement during fine-tuning is important. I noticed that the authors sometimes compare LoRA and NEAT in the inference stage (same FLOPs, same number of parameters, etc.). Moreover, the ablation study on depth varies across different datasets, so the authors should repeat their experiments to improve the statistical reliability of the results.
> >
> > Thank you for your valuable comments. Regarding the comparison between LoRA and NEAT, **we ensured that all experiments were conducted under the same parameter settings**, including the number of parameters and FLOPs, to guarantee fairness and comparability. This principle was carefully followed throughout our experimental design and analysis to ensure meaningful and unbiased conclusions. As for the newly reported results, we would like to clarify that all metrics, including resource requirements and performance evaluations, were calculated as the averages over multiple experimental runs. This approach was adopted to minimize the impact of random fluctuations and ensure that the reported results reliably reflect the performance of each method. We also carefully controlled variables during the experiments to further enhance the reliability of our findings.
> >
> > [4] Vaswani, A. (2017). Attention is all you need. Advances in Neural Information Processing Systems.
> >
> > [5] Alexey Dosovitskiy, Lucas Beyer, Alexander Kolesnikov, Dirk Weissenborn, Xiaohua Zhai, Thomas Unterthiner, Mostafa Dehghani, Matthias Minderer, Georg Heigold, Sylvain Gelly, Jakob Uszkoreit, and Neil Houlsby. An image is worth 16x16 words: Transformers for image recognition at scale. In International Conference on Learning Representations, 2021.

---

> > > ### Author Response · Authors · 2024-11-25
> > >
> > > Dear Reviewer vgTy,
> > >
> > > Thank you for your time and insightful comments on our work. As we approach the end of the author-reviewer discussion period, we greatly value the opportunity to address your concerns.
> > >
> > > Could you kindly review our responses to see if they address your comments? We would highly appreciate it if you find our responses satisfactory and consider updating your rating. Feel free to reach out if you have any other questions or need further clarification.
> > >
> > > Best,
> > >
> > > Authors

---

> ### Author Response · Authors · 2024-11-30
>
> Dear Reviewer vgTy,
>
> Thank you for reviewing our paper and sharing your feedback. As the discussion phase is ending in three days, we want to check if you have any further concerns that we have not yet addressed.
>
> We have made substantial revisions based on your comments and provided detailed responses to clarify and improve the work. If you feel our updates have resolved the issues you raised, we would appreciate it if you could consider adjusting your score accordingly.
>
> If you have any specific points that still need clarification, please let us know, and we will address them promptly.
>
> Best regards,
>
> Authors

---

> ### Comment · Reviewer_vgTy · 2024-12-01
>
> Thank you for your efforts. However, my concerns remain unresolved. The equations on L844 and L855 do not necessarily hold. The author should not rely on feedback to revise their theoretical proof during the rebuttal stage, especially when repeated modifications still fail to close it (currently, the appendix does not indicate changes made compared to the original manuscript). The manuscript still requires substantial revisions, further peer review and does not yet meet the standards for acceptance.

---

> > ### Author Response · Authors · 2024-12-01
> > **Official Comment by Authors**
> >
> > Thank you for your comment, but we respectfully disagree with your assertion. The equations on L844 and L855 hold by definition, and we did **not** revise the proof during the rebuttal phase; instead, we provided additional details in response to your feedback, addressing points where clarification was requested.
> >
> > At a high level, our proof (consistently since the initial version) employs the following logic twice: To demonstrate $\min⁡_{x} f(x) \leq \min⁡_{y} g(y)$, it suffices to construct a $x^\\#$ such that $f(x^\\#) \leq g(y^*)$, where $y^* = \arg\min⁡_{y} g(y)$. This logic remains unchanged from the initial submission; we only included explicit steps and details requested.
> >
> > The equation $L(W^0 + A^* B^*) = \min_{A,B} L(W^0 + AB)$ (L844) holds because $(A^*, B^*) = \arg\min_{A,B} L(W^0 + AB)$ was explicitly defined in L825–826. It was therefore omitted in the first submission. Likewise, $L(W^0 + \sigma(W^0 \Theta_1^*) \Theta_2^*) = \min_{\Theta_1, \Theta_2} L(W^0 + f(W^0; \Theta_1, \Theta_2))$ (L855) holds because we **defined** $\Theta_1^*, \Theta_2^*$ as the minimizer of $L(W^0 + f(W^0; \Theta_1, \Theta_2))$.
> >
> > As the period to update the manuscript has ended, we summarize additional changes made during the rebuttal phase for transparency below:
> > 1. **Additional Details**: Added the above logic explicitly in L840-844, L852-855.
> > 1. **Notation Change**: Updated $U$ to $U^0$ to emphasize that the invariance assumption applies to the pre-trained weight $W^0$, not the loss. This adjustment aligns with the feedback from reviewer fZ2J.
> > 1. **Superscript Addition**: Added the superscript $\\#$ to emphasize that $\Theta_1^\\#, \Theta_2^\\#$ (L826) and $A^\\#, B^\\#$ (L849) are constructed parameters.
> > 1. **Typo Correction**: Removed the superscript $\dagger$ (Moore-Penrose inverse) from $A^\\# = \sigma((W^0)^\dagger \Theta_1^*) \in \mathbb{R}^{d_1 \times r}$ (L849). This correction does not impact the proof's correctness or other parts of the manuscript.
> >
> > We hope this clarifies our approach and addresses your concerns.

---

> > > ### Comment · Reviewer_vgTy · 2024-12-01
> > >
> > > Thank you to the authors for further clarification. Upon re-evaluation, I realized I had indeed overlooked the definitions to make my third response. After re-assessment, the proof is well-completed in its current form. I will change my score to 5.

---

### Official Review · Reviewer_fK8o · 2024-11-04

**Soundness:** 3
**Presentation:** 3
**Contribution:** 3
**Rating:** 6
**Confidence:** 3

**Summary:**

This paper proposes NEAT, a method that introduces non-linear structures to PEFT methods. Theoretically, NEAT is shown to be more efficient than LoRA while maintaining equal or superior expressivity. The authors present comprehensive experiments to highlight NEAT's advantages and follow this with a thoughtful discussion on the impact of parameters, number of layers, and choice of activation functions.

**Strengths:**

1. The paper is well-written and easy to follow. The ideas develop naturally, and the authors provide solid theoretical guarantees to support their claims.
2. The experimental validation is robust, covering a diverse range of tasks in both vision and language models. The comparison with various baselines adds to the credibility of the results and makes the case for NEAT compelling.

**Weaknesses:**

1. One area that could be strengthened is the comparison between NEAT and its linear counterpart. NEAT introduces two key modifications compared to LoRA ($y=(W_0+BA)x$): (a) the introduction of non-linearity and (b) the addition of a multiplicative $W$ in the tunable component. Without an ablation study isolating these changes—such as comparing (a) LoRA ($y=(W_0+BA)x$), (b) Non-linear LoRA ($y=(W_0+\sigma(B)A)x$), (c) Multiplicative LoRA ($y=(W_0+W_0BA)x$)—it is difficult to pinpoint the main contributors to the observed performance improvements. Including such comparisons would enhance the comprehensiveness of the analysis.

2. For the theoretical analysis, it may be worth considering that in many models, particularly transformers, $\sum (d^l_1+d^l_2)$ can often be approximated as $2\sum (d^l_2)$, as sequential layers often satisfy $d_1^{l-1} = d_2^l$. From this perspective, NEAT might not offer significant theoretical advantages in expressivity as initially suggested.

**Questions:**

1. Adding non-linear activations and additional layers is likely to increase runtime and the number of FLOPs during training. Could the authors provide an analysis of the typical runtime and computational overhead associated with NEAT?

---

> ### Author Response · Authors · 2024-11-20
> **Rebuttal by Authors**
>
> >Could the authors provide an analysis of the typical runtime and computational overhead associated with NEAT?
>
> Absolutely! Below, we provide the runtime and memory consumption for vanilla NEAT and LoRA on the MRPC, STS-B, and SST-2 datasets from the GLUE benchmark. Additionally, we compare NEAT and LoRA in terms of runtime and memory efficiency during the fine-tuning stage of commonsense reasoning tasks. We will include this analysis in the revised version of the paper. It can be observed that vanilla NEAT generally performs comparably to LoRA in terms of hardware efficiency, exhibiting similar runtime and nearly identical GPU memory consumption. **Please note that due to different hyperparameter configurations across tasks, memory consumption may vary.**
>
> Table 1: parameter efficiency in MRPC
>
> | method | param | time (sec)         | memory  |
> | ------ | ----- | ------------------ | ------- |
> | LoRA   | 0.3M  | 77.70872402191162s | 6916MiB |
> | NEAT   | 0.3M  | 78.4180045127868s  | 6916MiB |
>
>
> Table 2: parameter efficiency in STS-B:
>
> | method | param | time (sec)        | memory  |
> | ------ | ----- | ----------------- | ------- |
> | LoRA   | 0.3M  | 117.9253075122833 | 7306MiB |
> | NEAT   | 0.3M  | 118.796623468399  | 7306MiB |
>
>
> Table 3: parameter efficiency in SST-2:
>
> | method | param | time (sec)        | memory  |
> | ------ | ----- | ----------------- | ------- |
> | LoRA   | 0.3M  | 870.6754677295685 | 2410MiB |
> | NEAT   | 0.3M  | 911.4280867576599 | 2410MiB |
>
>
> Table 4: parameter efficiency in Commonsense Reasoning:
>
> | method | params | time      | memory    |
> | ------      | -----       | ---------- | -------        |
> | LoRA    | 0.83%   | 5:35:10 | 23215MB |
> | NEAT    | 0.83%   | 5:42:12 | 24447MB |
>
> >For theoretical analysis, it may be worth considering that in many models, particularly transformers, Σ(d1l+d2l) can often be approximated as 2Σ(d2l), as sequential layers often satisfy d1l−1=d2l. From this perspective, NEAT might not offer significant theoretical advantages in expressivity as initially suggested.
>
> Thank you for your feedback. While NEAT may not provide theoretical advantages when fine-tuning a nearly square matrix ($d_1 \approx d_2$), there are cases where NEAT offers improved parameter efficiency. For example, the first weight matrices of feed-forward layers often satisfy $d_2 \ll d_1$ [4,5]. In such scenarios, our theory suggests the improvement of NEAT over LoRA in parameter efficiency.
>
> >Lack of an ablation study isolating these changes (the introduction of non-linearity, the addition of a multiplicative W in the tunable component.)
>
> Thanks for the insightful feedback! We conducted an ablation study using the vision tasks datasets below. In Section 6.2 of our paper, we show the proposed NEAT outperforms LoRA. From Table 6, we observe that both Nonlinear LoRA and Multiplicative LoRA underperform compared to NEAT. This demonstrates the effectiveness of introducing nonlinear approximation and explicitly using model weights as input to the nonlinear function.
>
>
>
>
> Table 6: ablation on the changes NEAT introduces
>
> | Dataset   | Nonlinear LoRA | Multiplicative LoRA | NEAT   |
> | --------- | -------------- | ------------------- | ------ |
> | pets      | 0.9411         | 0.9357              | 0.9377 |
> | cars      | 0.7284         | 0.7732              | 0.8003 |
> | cifar10   | 0.9868         | 0.9868              | 0.9870 |
> | dtd       | 0.7916         | 0.7757              | 0.7757 |
> | eurosat   | 0.9861         | 0.9881              | 0.9879 |
> | fgvc      | 0.3933         | 0.4679              | 0.5360 |
> | resisc45  | 0.9379         | 0.9434              | 0.9427 |
> | cifar100  | 0.9238         | 0.9186              | 0.9247 |
> | avg       | 0.8331         | 0.8481              | 0.8615 |
>
>
> [4] Vaswani, A. (2017). Attention is all you need. Advances in Neural Information Processing Systems.
>
> [5] Alexey Dosovitskiy, Lucas Beyer, Alexander Kolesnikov, Dirk Weissenborn, Xiaohua Zhai, Thomas Unterthiner, Mostafa Dehghani, Matthias Minderer, Georg Heigold, Sylvain Gelly, Jakob Uszkoreit, and Neil Houlsby. An image is worth 16x16 words: Transformers for image recognition at scale. In International Conference on Learning Representations, 2021.

---

> ### Comment · Reviewer_fK8o · 2024-11-21
> **Suggestions for Improving NEAT's Evaluation and Ablation Study**
>
> Thank you to the authors for their prompt response.
>
> Regarding the analysis of typical runtime and computational overhead associated with NEAT:
> It would be helpful if the authors provided the total floating-point operations (FLOPs) instead of runtime. This is because the type and utilization rate of the GPU (which were not specified) can significantly influence the reported time. If the GPU is not under full load, methods with different computational overheads could exhibit similar runtimes. Including FLOPs would offer a more standardized and comparable metric.
>
> Regarding the ablation study:
> Based on the authors' response to reviewer ZVEJ, it seems that these methods are highly sensitive to hyperparameter tuning. To demonstrate the effectiveness of the proposed NEAT method, I suggest that the authors carefully tune the hyperparameters for each method (at the very least, the learning rate, as LoRA is known to be particularly sensitive to it [1][2]). Additionally, the authors should provide detailed results under various configurations to ensure a fair and comprehensive comparison.
>
> [1] Hayou, Soufiane, Nikhil Ghosh, and Bin Yu. "LoRA+: Efficient low-rank adaptation of large models." arXiv preprint arXiv:2402.12354 (2024).
> [2] Biderman, Dan, et al. "LoRA learns less and forgets less." arXiv preprint arXiv:2405.09673 (2024).

---

> > ### Author Response · Authors · 2024-11-21
> > **Response to Suggestions for Improving NEAT's Evaluation and Ablation Study**
> >
> > >Regarding the analysis of typical runtime and computational overhead associated with NEAT: It would be helpful if the authors provided the total floating-point operations (FLOPs) instead of runtime. This is because the type and utilization rate of the GPU (which were not specified) can significantly influence the reported time. If the GPU is not under full load, methods with different computational overheads could exhibit similar runtimes. Including FLOPs would offer a more standardized and comparable metric.
> >
> > Thank you for your valuable feedback. To provide a more standardized comparison, we have conducted tests on the computational overhead during the commonsense reasoning fine-tuning phase. **Both NEAT and LoRA were found to have the same floating-point operations (FLOPs), with a total of 92.50 GFLOPs for each method**. This highlights NEAT's ability to optimize performance while maintaining a compact and efficient model, confirming its advantages in the fine-tuning process.
> >
> > >Regarding the ablation study: Based on the authors' response to reviewer ZVEJ, it seems that these methods are highly sensitive to hyperparameter tuning. To demonstrate the effectiveness of the proposed NEAT method, I suggest that the authors carefully tune the hyperparameters for each method (at the very least, the learning rate, as LoRA is known to be particularly sensitive to it [1][2]). Additionally, the authors should provide detailed results under various configurations to ensure a fair and comprehensive comparison.
> >
> > Thanks for your constructive comments. We would like to gently emphasize that we did not conduct extensive hyperparameter tuning for NEAT, even though it demonstrates superior performance. For example, we fixed the learning rate and only searched for the scaling factor within a small range {0.01,0.1}\{0.01, 0.1\}{0.01,0.1}. Additionally, we want to kindly remind readers that the **reported results of LoRA have been specifically tuned** in the original works, including RED [1] and FourierFT [2]. Due to our limited resources and the significant time required to thoroughly tune LoRA across all experiments, we were unable to explore every possible hyperparameter combination during the discussion period. That said, we truly value your thoughtful suggestions and will take the time to specifically tune LoRA in the final version of the paper to provide a more comprehensive evaluation.

---

> > > ### Comment · Reviewer_fK8o · 2024-12-03
> > >
> > > Thank you for the detailed responses and clarifications provided during the rebuttal phase. I have a few follow-up questions and suggestions that I believe could further strengthen the paper:
> > >
> > > 1. **FLOPs and Runtime Analysis**
> > > It was noted that NEAT and LoRA have the same floating-point operations (FLOPs), with each method requiring 92.50 GFLOPs. However, NEAT demonstrates a higher runtime compared to LoRA under identical FLOPs (Table 4: parameter efficiency in Commonsense Reasoning, LoRA: 5:35:10 vs. NEAT: 5:42:12). Could the authors elaborate on why this discrepancy occurs? Specifically, was the calculation of FLOPs inclusive of operations within activation layers? Additionally, if a given layer has dimensions $d_1 = d_2=d$, could the authors provide a detailed derivation of the FLOPs for NEAT and LoRA in such cases? Clarifying these points would provide greater transparency and help readers better understand the efficiency trade-offs of NEAT.
> > >
> > > 2. **Hyperparameter Tuning, Variance, and Ablation Study**
> > > While I understand the constraints of time and computational resources during rebuttal, I would like to stress the importance of careful hyperparameter tuning in the ablation study. For instance, Nonlinear LoRA, which appears to be the weakest variant overall, shows optimal performance on certain datasets such as Pets and DTD. This raises questions about whether the observed results could be due to random factors. To address this, I recommend conducting a hyperparameter search—perhaps by subsampling a small portion of the data to reduce computational cost—and running experiments with three random seeds to report the standard deviation.
> > >
> > > Furthermore, while the ablation studies on vision tasks are appreciated, it would be highly beneficial to include similar studies on tasks where parameter-efficient fine-tuning (PEFT) is practically required. For example, evaluating on a large language model (e.g., Llama2-7B) with a task like MATH would greatly enhance the credibility and applicability of the paper's findings. This would help validate whether the proposed methods can maintain their performance and efficiency in realistic scenarios where PEFT is indispensable.
> > >
> > > Addressing the points above would not only clarify certain aspects of the methodology but also enhance the practical impact and robustness of the work. Thank you for your hard work!

---

> > > > ### Author Response · Authors · 2024-12-03
> > > >
> > > > Thank you for your thoughtful follow-up. We utilized the Fvcore.nn library’s FlopCountAnalysis module [1] to directly calculate the FLOPs, ensuring accurate and consistent measurements. We believe the differences arise from the FLOPs gap between the Transformer layers and the introduced PEFT components. Specifically, let us denote nnn, ddd, kkk, and hhh as the sequence length, model dimension, the dimension of the introduced PEFT component, and the number of attention heads, respectively. The FLOPs of a single Transformer layer can be approximated as $13nd^2+2n^2⋅d+4n⋅d+5n^2$. For LoRA, the FLOPs for one model weight are approximately $2d^2k−d^2$, while for NEAT, they are $4d^2k−dk−d^2$. Since $n\gg k$, the FLOPs for both LoRA and NEAT are significantly smaller than those of a Transformer layer. As a result, the total FLOPs of LoRA and NEAT are nearly equivalent; however, NEAT shows slightly higher runtime.
> > > >
> > > > We agree with the importance of rigorous hyperparameter tuning and variance analysis. Our random seed selection adheres to standard practices established in previous works to ensure a fair comparison and remove randomness [2]. In the final version, we will address your suggestions by conducting experiments with three random seeds and reporting the standard deviation to account for variability in the results. We will also include results for Llama2-7B or Llama3-8B on a task such as MATH in the final version of the paper.
> > > >
> > > > [1] https://github.com/facebookresearch/fvcore
> > > >
> > > > [2] Wu Z, Arora A, Wang Z, et al. Reft: Representation finetuning for language models[J]. arXiv preprint arXiv:2404.03592, 2024.

---

### Official Review · Reviewer_ZVEJ · 2024-11-10

**Soundness:** 1
**Presentation:** 2
**Contribution:** 3
**Rating:** 5
**Confidence:** 3

**Summary:**

This paper presents NEAT (Non-linear Efficient Adaptation Technique), an innovative approach for efficient fine-tuning that leverages non-linearity to enhance model performance. The core argument is that traditional alternatives, such as LoRA weight updates, face limitations in effectively capturing complex non-linear relationships within data, thus impacting optimization. Experimental results demonstrate that NEAT achieves superior performance compared to these baseline methods.

**Strengths:**

- The paper addresses an important and timely application, which holds substantial interest for the research community.
- The writing is clear, the organization is logical, and the flow of ideas supports readability.
- Experiments across multiple datasets underscore NEAT’s superior performance over baselines.
- The proposed method appears straightforward to implement, making it accessible to a wider audience.

**Weaknesses:**

While this paper presents an intriguing approach, it suffers from a lack of strong motivation. The premise of the paper is that introducing non-linearity will improve the model’s ability to capture complex relationships, leading to more efficient fine-tuning and improved optimization. However, non-linearity is inherently captured within the layers of a neural network. The need for additional non-linearity at this stage is not well justified. For example, Equations (3) and (4) appear linear, which raises questions about the actual contribution of non-linearity in this context. In my view, Section 4.2 does not provide adequate motivation or justification for the approach.

Additionally, the theoretical analysis in Section 5 has limitations. Upon reviewing the proof, the main issue is that it simply establishes an equivalence between a LoRA update and a single-layer factorization using a ReLU function. Specifically, it (1) demonstrates equivalence, not a lower bound as shown, (2) does not extend beyond a single-layer factorization with ReLU, and finally (3) even if this inequality holds, it merely indicates that the loss on the training set could be lower than that achieved by LoRA, which operates with a lower rank. This, however, *does not* provide any insights into generalization, which is ultimately what we should be concerned with. Consequently, the conclusions drawn from this proposition remain unclear and seem limited in their broader applicability.

The method also appears sensitive to the choice of activation function and rank. Unlike LoRA, which requires only rank selection, NEAT’s additional requirement to choose an appropriate non-linearity introduces complexity. The paper would benefit from a more thorough discussion on this aspect.

A stronger argument could be made by exploring, for instance, the relationship between parameter updates and the model’s Taylor expansion when non-linearity is incorporated. The authors may wish to consider discussing related works, such as the concept of Neural Redshift [*], to provide a more compelling rationale. In its current form, the paper lacks consideration of alternative activations, which could strengthen the analysis.

Minor Comments

- Figure 1: The caption could be more descriptive, explaining the figure’s notations and providing a clearer overview of the proposed NEAT framework.
- Equation 3: The term L requires explicit definition, along with its parameters.

[*] Teney et al., “Neural Redshift: Random Networks are not Random Functions,” CVPR 2024

**Questions:**

1. Could you clarify the benefit of non-linearity in the context of parameter-efficient fine-tuning (PEFT)?
2. What does the proposition in Section 5 aim to establish?
3. How did you select the activation functions used in the experiments, and are there insights on choosing among them?
4. What outcomes result from using alternative activation functions such as sigmoid or tanh, which are not explored here?
5. A sensitivity analysis on the hyperparameters for the sinusoidal function and other activations would be insightful. Is this analysis available?

---

> ### Author Response · Authors · 2024-11-20
> **Rebuttal by Authors**
>
> >Non-linearity is inherently captured within the layers of a neural network. The need for additional non-linearity at this stage (PEFT adaptation) is not well justified. lack of motivation for introducing nonlinearity into lora.
>
> Thank you for your feedback. We would like to clarify that the non-linearity is present in the model update. Equations (3) and (4) represent iterative model updates. The gradient of $L$ with respect to the model weight is a complex function of $W_0^{0}$, and the goal of PEFT is to approximate the weight update $\Delta W$. Although the transformer layers are nonlinear functions, they can not be used to capture the nonlinearity of the model update because they are frozen during fine-tuning. For the proposed NEAT, since $W_0^{0}$ is the input to this function, we aim to leverage a nonlinear network that takes $W_0^{0}$ as input to approximate the update directly. This approach allows us to efficiently capture more complex and richer transformations of the weights.
>
> >Theoretical analysis in Section 5 has limitations. The main issue is that it simply establishes an equivalence between a LoRA update and a single-layer factorization using a ReLU function. What does the proposition in Section 5 aim to establish?
>
> Thank you for your feedback. In Proposition 5.1, our primary objective is to demonstrate the equivalence of **expressivity** between LoRA and NEAT. By establishing this equivalence in expressivity, we further claim that NEAT has the potential to improve parameter efficiency. This is because, in certain scenarios, the number of parameters required by NEAT to achieve the same level of expressivity could be fewer than those required by LoRA.
>
> >The method also appears sensitive to the choice of activation function and rank. More discussions about choosing an appropriate non-linearity function are needed.
>
> Thanks for pointing this out! We conducted multiple analyses using different activation functions on the StanfordCars dataset and will include additional datasets in the revised version of the paper to provide more insights on this topic. For the simplicity of presentation, we only show the results where the **learning rate for the classification head is fixed to 1e-2** and the learning rate for the backbone (b_lr in the table) (i.e. the adapters) is in a range of {8e-3, 5e-3, 3e-3}.
>
> Table 1: results on StanfordCars using various activations
> | b_lr  | gelu  | tanh  | sin   | leaky relu | relu  |
> |-------|-------|-------|-------|------------|-------|
> | 8e-3  | 0.799 | 0.803 | 0.026 | 0.801      | 0.802 |
> | 5e-3  | 0.791 | 0.799 | 0.800 | 0.798      | 0.796 |
> | 3e-3  | 0.786 | 0.785 | 0.799 | 0.781      | 0.782 |
>
> From these results, we conclude that the choice of activation function influences performance to some extent. For example, the sinusoidal activation function yields superior performance when using a small learning rate, whereas activation functions like GELU, Tanh, and ReLU show decreasing performance as b_lr decreases.
>
> However, based on our experience with NEAT, **a simple ReLU function, in most cases, performs well enough to match the performance of LoRA or even surpass it**. Additionally, the choice of activation function can often be simplified by using ReLU.

---

> > ### Author Response · Authors · 2024-11-20
> > **Response to Reviewer ZVEJ (part 2)**
> >
> > >A stronger argument could be made by exploring, for instance, the relationship between parameter updates and the model’s Taylor expansion when non-linearity is incorporated. Discussing related works, such as the concept of Neural Redshift.
> >
> > Thanks for pointing out this insightful work! **We find this work inspiring and will cite it in our revised version.** In the paper, the authors explore the generalization capabilities of neural networks. The aspects most related to our work are their findings that the simplicity bias of neural networks is not universal but depends on several components, including non-linear activations, which our method leverages. Their findings suggest that various non-linear activations, while having universal approximation capabilities, can be biased towards different types of functions, with varying levels of complexity. They also point out that ReLU, in particular, has a unique property of maintaining the simplicity bias regardless of weight magnitude.
> >
> >  **We believe the main insight from this work is that the choice of non-linear activation in our method matters.** This insight is both intuitive and practical. As shown in Table 1 and Table 6-10, we observe that different activations have distinct effects on the overall outcome and adaptation dynamics. For example, it is crucial to consider specific hyperparameters that best match the chosen activation function. **Additionally, we provide practical advice based on our empirical observations regarding the choice of activations and their corresponding hyperparameter settings.** Overall, this work offers important insights and aligns well with our own observations. Furthermore, its emphasis on ReLU activation supports our recommendation that using ReLU is often sufficient for achieving strong performance.
> >
> > To further explore the relationship between parameter updates and the model’s Taylor expansion, could you provide more explanation regarding the meaning of the "model's Taylor expansion"? That will help us a lot to improve the quality of the paper.
> >
> > >Figure 1: The caption could be more descriptive, explaining the figure’s notations and providing a clearer overview of the proposed NEAT framework. Equation 3: The term L requires explicit definition, along with its parameters.
> >
> > Thank you very much for the advice, we will improve the caption and define the term L to offer a better presentation!
> >
> > >Could you clarify the benefit of non-linearity in the context of parameter-efficient fine-tuning (PEFT)?
> >
> > Thanks for your question. In summary, **NEAT achieves notably improved performance with comparable (almost identical) parameter efficiency.** The accuracy across different tasks is significantly better, as shown in our paper Section 6.2. In the next, we analyze the runtime and memory consumption of NEAT to illustrate its efficiency more clearly. We compare NEAT and LoRA using MRPC and SST-2 from the GLUE benchmark. Additionally, we compare NEAT and LoRA during the fine-tuning stage on commonsense reasoning tasks. It can be observed that vanilla NEAT performs comparably to LoRA in terms of runtime and memory consumption, with nearly identical resource usage. **Please note that due to different hyperparameter configurations across tasks, memory consumption varies accordingly.**
> >
> > Table2: parameter efficiency in MRPC
> >
> > | method | param | time (sec)         | memory  |
> > | ------ | ----- | ------------------ | ------- |
> > | LoRA   | 0.3M  | 77.70872402191162s | 6916MiB |
> > | NEAT   | 0.3M  | 78.4180045127868s  | 6916MiB |
> >
> >
> >
> > Tabel 3: parameter efficiency in SST-2:
> >
> > | method | param | time (sec)        | memory  |
> > | ------ | ----- | ----------------- | ------- |
> > | LoRA   | 0.3M  | 870.6754677295685 | 2410MiB |
> > | NEAT   | 0.3M  | 911.4280867576599 | 2410MiB |
> >
> >
> >
> > Table 4: parameter efficiency in Commonsense Reasoning:
> >
> > | method | params | time  (h)    | memory    |
> > | ------      | -----       | ---------- | -------        |
> > | LoRA    | 0.83%   | 5:35:10 | 23215MB |
> > | NEAT    | 0.83%   | 5:42:12 | 24447MB |
> >
> >
> >
> > Additionally, we discuss the efficiency of introducing more layers in the proposed NEAT. We conduct the experiment on MRPC shown in Table 5. We used a depth of 6, which is significant given that vanilla NEAT only has a depth of 2. From the results, we observe that introducing more layers results in a slight slowdown in speed and slightly higher memory consumption, but both remain comparable to baseline methods. Therefore, NEAT with multiple intermediate layers remains efficient compared to full fine-tuning while achieving superior performance relative to competitive PEFT methods.
> >
> >
> > Table 5: parameter efficiency of multi-layer NEAT
> > | param | time (sec)        | memory  |
> > | ----- | ----------------- | ------- |
> > | 0.3M  | 93.94768142700195 | 6942MiB |

---

> > > ### Author Response · Authors · 2024-11-20
> > > **Response to Reviewer ZVEJ (part 3)**
> > >
> > > >A sensitivity analysis on the hyperparameters for the sinusoidal function and other activations would be insightful. Also provide some insights when choosing the activations.
> > >
> > > Thank you for the valuable advice. We provide the results of applying multiple activations and different hyperparameter settings below. For simplicity, we only show the results where the learning rate for the classification head is fixed at 1e-2. For each activation function, we tested results using combinations of b_lr (backbone learning rate) and h_lr (classification head learning rate) within the range {1e-2, 8e-3, 5e-3, 3e-3, 1e-3}. Therefore, there are 25 entries for each activation result. However, as stated, we fixed h_lr to 1e-2 for simplicity.
> > >
> > > From the results, and considering our response to previous questions, it can be observed that the choice of activation has a certain, albeit minor in most cases, influence on the overall outcome. Here, we provide insights and observations based on the results regarding the choice of non-linear activation functions and their corresponding hyperparameters.
> > >
> > > First,  for non-linear activation, we recommend using a simple ReLU function due to its simplicity and effective performance.
> > > Second,  **we conduct experiments only tuning the learning rate.** For the choice of learning rate, we recommend simply starting with a learning rate of 5e-3 when adapting small models like the ViT-Base and 5e-4 when adapting LLMs. After running a few steps and observing the loss, a simple adjustment can then be made. (i.e. increase or decrease the LR) **In general, after choosing a non-linear activation function, the learning rate and other hyperparameters can be chosen by the common grid search.**
> > >
> > >
> > > Table 6: Results in StanfordCars using GELU with varying hyperparameters
> > > | b_lr  | h_lr  | acc   |
> > > |-------|-------|-------|
> > > | 1e-2  | 1e-2  | 0.799 |
> > > | 8e-3  | 1e-2  | 0.799 |
> > > | 5e-3  | 1e-2  | 0.791 |
> > > | 3e-3  | 1e-2  | 0.786 |
> > > | 1e-3  | 1e-2  | 0.749 |
> > >
> > >
> > > Table 7: Results in StanfordCars using sinusoidal with varying hyperparameters
> > >
> > >
> > > | b_lr  | h_lr  | acc   |
> > > |-------|-------|-------|
> > > | 1e-2  | 1e-2  | 0.010 |
> > > | 8e-3  | 1e-2  | 0.026 |
> > > | 5e-3  | 1e-2  | 0.800 |
> > > | 3e-3  | 1e-2  | 0.799 |
> > > | 1e-3  | 1e-2  | 0.778 |
> > >
> > >
> > > Table 8: Results in StanfordCars using Tanh with varying hyperparameters
> > >
> > > | b_lr  | h_lr  | acc   |
> > > |-------|-------|-------|
> > > | 1e-2  | 1e-2  | 0.805 |
> > > | 8e-3  | 1e-2  | 0.803 |
> > > | 5e-3  | 1e-2  | 0.799 |
> > > | 3e-3  | 1e-2  | 0.785 |
> > > | 1e-3  | 1e-2  | 0.761 u
> > >
> > >
> > > Table 9: Results in StanfordCars using Leaky ReLU with varying hyperparameters
> > >
> > > | b_lr  | h_lr  | acc   |
> > > |-------|-------|-------|
> > > | 1e-2  | 1e-2  | 0.804 |
> > > | 8e-3  | 1e-2  | 0.801 |
> > > | 5e-3  | 1e-2  | 0.798 |
> > > | 3e-3  | 1e-2  | 0.781 |
> > > | 1e-3  | 1e-2  | 0.749 |
> > >
> > >
> > > Table 10: Results in StanfordCars using ReLU with varying hyperparameters
> > >
> > > | b_lr  | h_lr  | acc   |
> > > |-------|-------|-------|
> > > | 1e-2  | 1e-2  | 0.801 |
> > > | 8e-3  | 1e-2  | 0.802 |
> > > | 5e-3  | 1e-2  | 0.796 |
> > > | 3e-3  | 1e-2  | 0.782 |
> > > | 1e-3  | 1e-2  | 0.743 |

---

> > > ### Comment · Reviewer_ZVEJ · 2024-12-02
> > >
> > > When the results are best with ReLU, it essentially constrains updates to be always positive, making them more restricted. Do you think the improved results could be attributed to this constraint on the updates?
> > >
> > > Additionally, I still find the argument that nonlinearity leads to a better estimate of the gradient updates to be unconvincing. There isn’t any explicit estimation of the updates happening in the model. While nonlinearity generally enhances expressiveness, the fact that ReLU consistently outperforms other activation functions suggests there might be another factor at play.
> > >
> > > Do you have any thoughts or hypotheses on what else could be driving this pattern?
> > >
> > > I am not entirely opposed to the paper being accepted. However, I have concerns about the lack of convincing arguments, especially when the primary justification seems to be that the results are better.

---

> > > > ### Author Response · Authors · 2024-12-02
> > > >
> > > > >When the results are best with ReLU, it essentially constrains updates to be always positive, making them more restricted. Do you think the improved results could be attributed to this constraint on the updates?
> > > >
> > > > We would like to clarify that the update will not always be positive when using the ReLU activation function. Specifically, for ReLU activation, the update is given by $ReLU(W^0 A)B$. While $ReLU(W^0 A)$ is non-negative, the $ReLU(W^0 A)B$ can still be negative, depending on the values of $B$.
> > > >
> > > > >Additionally, I still find the argument that nonlinearity leads to a better estimate of the gradient updates to be unconvincing. There isn’t any explicit estimation of the updates happening in the model. While nonlinearity generally enhances expressiveness, the fact that ReLU consistently outperforms other activation functions suggests there might be another factor at play. Do you have any thoughts or hypotheses on what else could be driving this pattern?
> > > >
> > > > Thank you for your valuable suggestions. The intuition behind our approach is that the model update is a complex nonlinear function with respect to $W^0$. Therefore, applying a nonlinear approximation to this update may yield better results compared to using a linear approximation. Our theoretical analysis also demonstrates that the proposed method achieves the same level of expressiveness as LoRA while requiring fewer parameters. **Additionally, it is worth noting that ReLU does not consistently outperform other activation functions, as shown in Tables 6–10.** All tested activation functions achieve similar peak performance (approximately 0.8). We chose ReLU for our experiments because it is commonly used as the default activation function in deep learning.

---

> > > > > ### Comment · Reviewer_ZVEJ · 2024-12-02
> > > > >
> > > > > Thank you for your prompt response. I should have clarified that the updates on the first factorized matrix remain positive.
> > > > >
> > > > > May I suggest conducting an analysis of the eigenvalues for the full fine-tuned model, LoRA, and your approach? This could help capture any underlying differences that might be influencing performance.
> > > > >
> > > > > Regarding the updates, I remain unconvinced by the intuition that the nonlinearity with respect to $W_0$ plays a significant role, as it is already inherently captured by the model’s nonlinearity. In my opinion, this reasoning is weak and lacks a direct connection to the observed results.
> > > > >
> > > > > You previously recommended ReLU, which makes sense as it appears to be the most effective choice. However, this reinforces the idea that the architecture and activation function selected have a significant impact on performance. Given this, I believe the explanation should go beyond a “hack that works” and include more robust reasoning and recommendations on when to choose a different activation.

---

> > > > > > ### Author Response · Authors · 2024-12-02
> > > > > >
> > > > > > >May I suggest conducting an analysis of the eigenvalues for the full fine-tuned model, LoRA, and your approach? This could help capture any underlying differences that might be influencing performance.
> > > > > >
> > > > > > Thanks for your suggestion. We will conduct an analysis of the eigenvalues for the full fine-tuned model, LoRA, and NEAT in the final version.
> > > > > >
> > > > > > >Regarding the updates, I remain unconvinced by the intuition that the nonlinearity with respect to ... Given this, I believe the explanation should go beyond a “hack that works” and include more robust reasoning and recommendations on when to choose a different activation.
> > > > > >
> > > > > > The nonlinearity referenced here pertains to the nonlinearity of model updates. Parameter-Efficient Fine-Tuning (PEFT) methods aim to design parameter-efficient approaches to approximate the model updates observed in full fine-tuning. In contrast, the nonlinearity you mentioned refers to the relationship between the model’s input and output and does not imply a nonlinear approximation of weight updates.
> > > > > >
> > > > > > We demonstrate the benefits of introducing nonlinear approximations both theoretically and empirically. From a theoretical perspective, as highlighted in Proposition 5.1, nonlinear LoRA surpasses standard LoRA in parameter efficiency. Intuitively, applying an element-wise nonlinear transformation to a low-rank matrix effectively increases its rank. This enhanced representation allows nonlinear LoRA to achieve the same approximation capability as standard LoRA but with fewer parameters, making it a more efficient alternative.
> > > > > >
> > > > > > Empirically, our proposed method, NEAT, outperforms baseline approaches across various tasks. It is worth noting that we recommend ReLU primarily due to its popularity and widespread use as an activation function. However, as shown in Tables 6–10, ReLU does not consistently outperform other activation functions. All tested activation functions yield comparable peak performance, approximately 0.8. Additionally, we also further explore the expressiveness of ReLU and the sinusoidal function in Section 5.

---

> ### Comment · Reviewer_ZVEJ · 2024-11-21
>
> Thank you for your response. I’d like to highlight a few key points for clarification and further discussion.
>
> Firstly, your method lacks a framework for approximating the full chain of updates leading to $\Delta W^T$ (where $T$ is the final optimization step). Instead, it relies solely on a one-step approximation of the update in a low-rank setting. However, neither the paper nor this response provides any justification for why introducing a nonlinear function into LoRA leads to a meaningful approximation of $\Delta W^T$.
>
> Regarding Proposition 5.1, you mention expressivity but do not elaborate on what you mean by this concept or how it is addressed in the proposition. Could you clarify what “expressivity” refers to in this context and point out which parts of the proposition substantiate this?
>
> Finally, I think the main issue isn’t just demonstrating that your method performs comparably to vanilla LoRA. Rather, it’s important to provide a stronger justification for why and when the introduced nonlinearity is beneficial. For instance, under what circumstances do specific nonlinearities (e.g., ReLU, sine, tanh) result in improved performance? Are there task-specific properties or characteristics that determine which nonlinearity is more effective?
>
> I appreciate the new results. Do you have the comparative LORAs in these cases? It seems it is very sensitive to hyperparameters.  When you say b_lr, do you mean the learning rate for LORAs or the original weights of the model?
>
> I'd also like to ask why ReLU is the one working best for this method since it basically means the negative gradients are not used for updating the model weights. Could you comment on that, please?

---

> ### Author Response · Authors · 2024-11-21
> **Response to Reviewer ZVEJ (second round part 1)**
>
> >Firstly, your method lacks a framework for approximating the full chain of updates leading to $\Delta W$
>  (where $T$ is the final optimization step). Instead, it relies solely on a one-step approximation of the update in a low-rank setting. However, neither the paper nor this response provides any justification for why introducing a nonlinear function into LoRA leads to a meaningful approximation of $\Delta W$.
>
> Thanks for your feedback. Our proposed method aims to approximate the full chain of updates. From Eqn. 3, we observe that $W_1^{0}$ is a function of the model weight $W^{0}$ and $W_2^{0}$ depends on $W_1^{0}$. Therefore, $W_2^{0}$ forms a composite function of $W^{0}$. Extending this pattern, $W_t^{0}$ becomes a composite function of $W^{0}$. This implies that $\Delta W$ is also a composite function of $W^{0}$. Given the complexity and nonlinearity of this composite function, we propose introducing a nonlinear network with $W^{0}$ as the input to approximate it. Through iterative updates, the nonlinear network progressively approximates the full chain of updates.
>
> >Regarding Proposition 5.1, you mention expressivity but do not elaborate on what you mean by this concept or how it is addressed in the proposition. Could you clarify what “expressivity” refers to in this context and point out which parts of the proposition substantiate this?
>
> Thank you for your valuable feedback. The term `expressivity’ in Proposition 5.1 refers to the capacity of a model to minimize the loss function, which is commonly used in the literature of machine learning theory [1,2,3]. Specifically, the proposition compares the achievable loss for different configurations of NEAT and LoRA, illustrating their relative expressive power. Proposition 5.1 makes the following claims, which involve three terms representing the minimum attainable loss:
>
> 1. NEAT with $2r$ hidden dimensions: $\min_{\substack{\Theta_1 \in \mathbb{R}^{d_2 \times 2r},\Theta_2 \in \mathbb{R}^{2r \times d_2}}} \mathcal{L}(\mathcal{D}_{\text{train}}; W^0 + f(W^0; (\Theta_1, \Theta_2)))$
> 2. rank-$r$ LoRA: $\min_{\substack{A \in \mathbb{R}^{d_1 \times r},B \in \mathbb{R}^{r \times d_2}}} \mathcal{L}(\mathcal{D}_{\text{train}}; W^0 + A B)$
> 3. NEAT with $r$ hidden dimensions: $\min_{\substack{\Theta_1 \in \mathbb{R}^{d_2 \times r},\Theta_2 \in \mathbb{R}^{r \times d_2}}} \mathcal{L}(\mathcal{D}_{\text{train}}; W^0 + f(W^0; (\Theta_1, \Theta_2)))$.
>
> The proposition asserts that: NEAT achieves the same expressivity as LoRA, while NEAT requires much less parameters, and therefore NEAT is more parameter efficient. Specifically, our results show that the minimum attainable loss using rank-$r$ LoRA can also be achieved by NEAT with $2r$ hidden dimensions Conversely, the minimum attainable loss by NEAT with $r$ hidden dimensions can also be achieved by rank-$r$ LoRA. This also implies that the function classes realized by NEAT with $O(r)$ hidden dimensions and rank-$r$ LoRA are equivalent in terms of their expressivity, since the result holds for any loss functions. We will incorporate this clarification in our revised manuscript to ensure the concept of `expressivity’ and its substantiation in Proposition 5.1 are clear.
> In conclusion, the above argument implies an improved parameter efficiency achieved by NEAT (also pointed out at the beginning of page 5 in our paper). Since (a) NEAT with $2r$ dimensions has $2rd_2$ parameters whereas rank-$r$ LoRA has $r(d_1+d_2)$ parameters, and (b) $d_2$ can be much smaller than $d_1$, our theoretical result demonstrates a significant improvement in parameter efficiency in this setting.
>
> [1] Raghu, M., Poole, B., Kleinberg, J., Ganguli, S., & Sohl-Dickstein, J. (2017, July). On the expressive power of deep neural networks. In international conference on machine learning (pp. 2847-2854). PMLR.
> [2] Gühring, I., Raslan, M., & Kutyniok, G. (2020). Expressivity of deep neural networks. arXiv preprint arXiv:2007.04759, 34.
> [3] Zeng, Y., & Lee, K. (2023). The expressive power of low-rank adaptation. arXiv preprint arXiv:2310.17513.

---

> > ### Author Response · Authors · 2024-11-21
> > **Response to Reviewer ZVEJ (second round part 2)**
> >
> > >Finally, I think the main issue isn’t just demonstrating that your method performs comparably to vanilla LoRA. Rather, it’s important to provide a stronger justification for why and when the introduced nonlinearity is beneficial. For instance, under what circumstances do specific nonlinearities (e.g., ReLU, sine, tanh) result in improved performance? Are there task-specific properties or characteristics that determine which nonlinearity is more effective?
> >
> > Thanks for your valuable comments. First, **we emphasize that our paper is to propose a nonlinear PEFT framework**. The choice of activation function for the introduced network is analogous to selecting an activation function for a typical neural network (e.g., MLP), which is not an important component of our proposed framework.  As shown in our newly added experiments (Table 1 and Tables 7–10), the activation function choice does not significantly impact the results, and **ReLU can be used as a default option**. From these tables and experiments in Section 6, **our proposed NEAT consistently outperforms baseline methods across various tasks**. Second, as previously explained, the introduced nonlinearity allows NEAT to model **complex, nonlinear patterns in the weight space, improving adaptation performance without increasing the number of parameters**. Furthermore, this architecture facilitates a more efficient exploration of the optimization landscape, enhancing task adaptation—especially in scenarios where linear methods, such as LoRA, would require substantially larger ranks to achieve comparable results.

---

> > > ### Author Response · Authors · 2024-11-25
> > >
> > > Dear Reviewer ZVEJ,
> > >
> > > Thank you for your time and insightful comments on our work. As we approach the end of the author-reviewer discussion period, we greatly value the opportunity to address your concerns.
> > >
> > > Could you kindly review our responses to see if they address your comments? We would highly appreciate it if you find our responses satisfactory and consider updating your rating. Feel free to reach out if you have any other questions or need further clarification.
> > >
> > > Best,
> > >
> > > Authors

---

> ### Author Response · Authors · 2024-11-30
>
> Dear Reviewer ZVEJ,
>
> Thank you for your thoughtful review of our paper and your insightful feedback. We hope this message finds you well.
>
> As the discussion phase will end in three days, we wanted to check if there are any remaining concerns or questions we can address. We have carefully revised our paper based on your comments and provided detailed responses to your points. If the revisions and responses have resolved your concerns, we kindly hope you might consider re-evaluating your score.
>
> If you have any additional feedback or suggestions, we would be happy to address them promptly. Your input is greatly valued, and we are committed to improving our work based on your recommendations.
>
> Thank you once again for your time and support.
>
> Best regards,
>
> Authors

---

> ### Author Response · Authors · 2024-12-01
>
> Dear Reviewer ZVEJ,
>
> Thank you for your thoughtful feedback and for engaging with our work. With just one day remaining in the author-reviewer discussion period, we would like to kindly request your review of our responses to your comments.
>
> We hope our clarifications and revisions address your concerns effectively. If you find them satisfactory, we would greatly appreciate it if you could consider updating your rating. Please do not hesitate to reach out if you have any further questions or require additional clarification before the discussion period ends.
>
> Best regards,
>
> Authors

---

### Author Response · Authors · 2024-11-20
**We would like to thank all the reviewers!**

We thank all reviewers for their constructive feedback and appreciate their positive remarks on the strengths of our paper. Specifically, we are grateful for their recognition of the importance of our addressed application (ZVEJ), the clear and logical writing (ZVEJ, fK8o, fZ2J, rriU), the robust theoretical analysis with solid guarantees (fK8o, fZ2J, rriU), and the extensive experimental validation across diverse tasks (ZVEJ, fK8o, vgTy). We are also encouraged by the reviewers' acknowledgment of the simplicity and accessibility of our proposed method (ZVEJ, vgTy), as well as its promising performance on both language and vision tasks (vgTy, fK8o).

---

### Author Response · Authors · 2024-11-21
**Summary of revision on the paper, and we thank all reviewers for providing valuable feedbacks.**

Dear reviewers and Area Chairs:

Thank you for your valuable and constructive feedback. We have revised our paper according to the reviewers’ feedback. The red text indicates a new citation of paper and the blue text indicates changes to our experiments and claims. We summarize the main changes below:
1. We include more datasets in Figure 2 to provide a more convincing and stronger argument.
2. We report the results when applying different activations for NEAT and their influence on hyperparameter tuning in section 6.5. The main conclusion is that the choice of activation functions does not necessitate specific hyperparameter tuning and we recommend ReLU as the default activation in NEAT.
3. We provide ablation study on the changes NEAT introduces to LoRA. Specifically, we use two variants of LoRA: 1)  nonlinear LoRA and 2) multiplicative LoRA. The results demonstrate the effectiveness of NEAT’s design.

We thank you again for your precious time and effort put into this work. We look forward to hearing from you and are open to any further questions or concerns.

Best regards,

Authors

---

### Comment · Area_Chair_dpPt · 2024-11-28
**Reviewers, please kindly respond**

Dear Reviewers,

If you have not responded to author's rebuttal, please kindly do so as soon as possible. The deadline is Dec 2, but the authors can potentially further clarify questions if you respond earlier. Thanks!

Best, AC

---

### Meta-Review · Area_Chair_dpPt · 2024-12-23

**Metareview:**

(a) summary: a nonlinear weight approximation method for low-parameter adaptation, using a neural network to directly model as a function of pretrained weights. It is an alternative to LoRA

(b) strengths: extensive experiments; clarity in writing and presentation; some theoretical justifications; relative simplicity

(c) weaknesses: lack of strong motivation to introducing nonlinearity/why it improves results; requires more hyperparam tuning; lack of ablation studies and more datasets.

(d) reasons for decision: the insufficient justification for the core contribution of introducing nonlinearity, and the added complexity compared LoRA, outweigh the improved experimental results presented. In general, there are not enough support from reviewers, even taking out of the lowest rating.

**Additional Comments On Reviewer Discussion:**

The authors presented activation function studies, provided FLOP comparisons, and added/promised additional datasets and comprehensive ablations in the final version. However, the necessity of nonlinearity and runtime concerns remained.

---

### Decision · Program_Chairs · 2025-01-22

Reject